# Recurrent circuits within medial entorhinal cortex superficial layers support grid cell firing

Ipshita Zutshi [1], Maylin L. Fu[1], Varoth Lilascharoen [1], Jill K. Leutgeb [1], Byung Kook Lim [1] & Stefan Leutgeb [1,2]

Specialized cells in the medial entorhinal cortex (mEC), such as speed cells, head direction (HD) cells, and grid cells, are thought to support spatial navigation. To determine whether these computations are dependent on local circuits, we record neuronal activity in mEC layers II and III and optogenetically perturb locally projecting layer II pyramidal cells. We find that sharply tuned HD cells are only weakly responsive while speed, broadly tuned HD cells, and grid cells show pronounced transient excitatory and inhibitory responses. During the brief period of feedback inhibition, there is a reduction in specifically grid accuracy, which is corrected as firing rates return to baseline. These results suggest that sharp HD cells are embedded in a separate mEC sub-network from broad HD cells, speed cells, and grid cells. Furthermore, grid tuning is not only dependent on local processing but also rapidly updated by HD, speed, or other afferent inputs to mEC.

[1] Neurobiology Section and Center for Neural Circuits and Behavior, Division of Biological Sciences, University of California, San Diego, La Jolla, CA, 92093, USA. [2] Kavli Institute for Brain and Mind, University of California, San Diego, La Jolla, CA, 92093, USA. Correspondence and requests for materials should be addressed to S.L. (email: sleutgeb@ucsd.edu)

The medial entorhinal cortex (mEC) harbors several functional cell types that are thought to be essential for spatial navigation and memory. These cell types include grid cells —cells that fire in striking hexagonally arranged fields[1,2], head direction (HD) cells—cells that fire only when an animal's head is facing a particular direction[3], and speed cells—cells whose firing rates are modulated by the running speed of an animal[4]. The co-localization of these functional cell types in the superficial layers (layers II and III) of mEC[4–7], along with the high proportion of grid cells within layer II of the mEC[2], has led to standard models of grid cell generation that require the integration of HD and speed information within local circuits as well as recurrent connectivity between grid cells[8–14]. While the neural circuit that forwards HD information from the anterior thalamic nucleus via the presubiculum to mEC is well described[15,16], the source of the speed signal to grid cells within the mEC remains less certain[17]. Speed information could either be derived from the frequency and amplitude modulation of theta oscillations by running speed[18] or from the readout of the firing rate of speed-modulated cells within mEC[4,19]. Despite the uncertainty about the source of speed information, HD and speed information have been proposed to be combined into a velocity signal before being forwarded to generate grid cells[8,12].

Although the site and mechanism for the processing and integration of speed and HD information remain unresolved, it is assumed that HD and speed signals are conveyed by specialized afferent pathways to mEC. Therefore, most investigations on grid generation have thus far focused on brain regions that strongly project directly and indirectly to the mEC. Accordingly, it has been demonstrated that afferent inputs from the hippocampus[20], the medial septum[21–23], and the anterior thalamic nucleus[16] are required for the periodic firing patterns of grid cells. These manipulations were found to have effects on spatial information, speed modulation, theta oscillations, directional tuning, or a combination thereof. Past findings are thus consistent with the general notion that a disruption in either heading or speed information blocks the neuronal computations required for grid firing. However, details on how each of the long-range input streams is combined within local networks remain to be identified. Unexpectedly, experiments that disrupted local circuits within mEC—one that targeted local parvalbumin (PV)-expressing interneurons[24] and the other that targeted stellate cells in layer II[25]—did not observe any effects on grid firing patterns. In addition, a recent study that inhibited mEC PV cells increased firing rates of grid cells predominantly outside of grid fields while grid centers remained aligned[26]. The limited effects of local circuit manipulations on grid cells therefore raise the possibility that dendritic processing or ion channel composition of a cell predominantly contribute to grid generation and that grid firing may thus selectively emerge in a particular morphological cell type. Numerous studies have therefore compared the two major morphological cell types in mEC layer II—stellate (LIIS) and pyramidal (LIIP) cells. The combined evidence from these studies suggests that grid cells can be found in either population[6,27–29]. Furthermore, altering cellular properties by knocking out HCN1 channels, which are most abundant in LIIS cells, did not interfere with the generation of grid patterns and only affected grid spacing[30]. Thus studies addressing either cellular or circuit computations within the mEC have not clearly determined whether local processing within the mEC superficial layers is required for sustaining grid firing patterns.

To address whether local circuits in the superficial layers support the firing patterns of functional cell types in mEC, we considered the detailed anatomical organization of connections within the superficial layers that were revealed by performing paired recordings in entorhinal slices (see Fig. 1a for a schematic).

While LIIP cells were with limited sampling reported to not project to LIIS cells[31], recent studies have shown dense direct projections from LIIP to LIIS cells, to other LIIP cells, and to local inhibitory interneurons[32,33]. Particularly remarkable is that synaptic connectivity from LIIP to LIIS exceeds a probability of 10% and is characterized by exceptionally strong and fast synaptic transmission[33]. Reports on the local projections of LIIS cells have also been conflicting[31,34,35], but large-scale sampling has confirmed that they neither extensively target LIIP nor LIII pyramidal cells but rather send dense projections to the deep layers of the mEC[33,36]. Based on these differences in connection patterns between LIIP and LIIS cells, we reasoned that manipulations of LIIP cells would be particularly effective in perturbing neuronal activity in the superficial layers. We therefore first confirmed that LIIP cells have the majority of their synaptic terminals within mEC superficial layers[36] and then optogenetically activated LIIP cells to recruit LIIS cell excitation, and as a consequence of direct and indirect activation of both principal cell types, interneuron driven feedback inhibition that transiently perturbed network activity across layers II and III. We then used this approach to test whether HD and speed tuning is disrupted together with grid firing.

## Results

**LII pyramidal cells project densely within LII of mEC.** To selectively perturb neuronal activity within the mEC superficial layers, we sought to manipulate a cell population that projects predominantly within the superficial layers and has limited synaptic contacts in other mEC layers and areas outside of mEC. LIIP cells have been shown to project within LII (Fig. 1a), while only sending a minor projection to CA1[32,33,36–38]. We therefore first examined the density of LIIP cell projections within the superficial layers of entorhinal cortex by using a transgenic mouse line that expresses tamoxifen-inducible Cre recombinase protein under the Wolfram syndrome 1 homolog promoter (Wfs1-creER mice). We confirmed the previously reported result that this line exclusively labels LIIP cells, with no overlap with the dentate gyrus (DG)-projecting LIIS cells[36] (Fig. 1b). To then compare the relative density of pyramidal cell projections to different targets, we injected a Cre-dependent adeno-associated virus (AAV) expressing mRuby2 and synaptophysin linked to enhanced green fluorescent protein (eGFP) (AAV-hSyn-FLEX-mRuby2-Syp-eGFP) into the mEC of Wfs1-creER mice (Fig. 1c) to label cell bodies and axons of LIIP cells with the soluble fluorophore mRuby2 (in red) and synaptic terminals with synaptophysin-eGFP (in green). Using this method, we confirmed not only the previously reported ipsilateral and contralateral projections within the mEC[32,33,36,38,39] but also identified that the ipsilateral projections were far denser (Fig. 1d, e) and had greater spread (mean ± sem, Ipsilateral: $0.2405 \pm 0.0223$ mm$^2$, Contralateral: $0.0176 \pm 0.0028$ mm$^2$. Wilcoxon matched-pairs signed-rank test, $n = 9$ sections from 3 mice, $W(9) = -45$, $p = 0.0039$) than the projections to the contralateral mEC. We found that the ipsilateral projections formed a network of synapses within LII where both patches of Wfs1+ cells and neighboring regions with Wfs1− cells appeared to be densely surrounded by synapses of LIIP cells (Fig. 1f). Outside of the mEC, we only observed the previously described projections to CA1 (Fig. 1g) and no projections to any other brain regions. Thus the dense local connectivity within layer II but not other mEC layers, along with the particularly weak projections to other brain areas, provided an ideal system to specifically manipulate the superficial layers of mEC.

**Functional connectivity of LIIP recurrent circuits.** To determine whether the dense ipsilateral projections observed with

anterograde viral tracing provided functional local control of neuronal activity in the intact brain, we optogenetically activated LIIP cells while recording single-unit activity from cells across LII and LIII in freely behaving mice. We began by expressing Cre-dependent Channelrhodopsin2 (AAV-EF1α-DIO-ChR2-eYFP) in the mEC of Wfs1-creER mice. Next, we implanted a movable array consisting of an optic fiber and four tetrodes (optetrode with the tetrodes <1 mm deeper than the tip of the optic fiber)[40], which permitted simultaneous single-unit recordings and optogenetic stimulation in the mEC (Fig. 2a). We then performed recordings of LII and LIII cells in an open field arena during baseline sessions without optical manipulations and during stimulation sessions. Throughout the entire stimulation session, LIIP cells were repetitively activated at one of the two stimulation frequencies—8 or 12 Hz. The two distinct frequencies were chosen to control for the possibility that there might be a frequency-dependent effect of rhythmic excitation of LIIP cells, especially for frequencies close to theta oscillations (~8 Hz). Moreover, both of these frequencies not only allowed sufficient time for recovery of cells between light pulses (83.3 ms for 12 Hz, 125 ms for 8 Hz) but also regularly perturbed the network throughout the session. We observed that most cells responded to optogenetic stimulation by increased excitation, synaptic inhibition, or both (Fig. 2b, Supplementary Fig. 2). As expected based on the strong excitatory circuits within LII[33], directly and synaptically activated cells could not be distinguished based on commonly used criteria such as latency or jitter[24,40,41] (Supplementary Fig. 2, also see Online Methods). Collectively, excitation was observed in a large fraction of LII cells (~50%) with the peak latency at ~6–7 ms after light onset.

Because LIIP cells were reported to not project to LIII[33], we analyzed whether LIII cells may be differently affected by the stimulation than LII cells. Consistent with the limited excitatory connectivity between LII and LIII, only ~12% of LIII cells were excited (Fig. 2c–e, Supplementary Table 1) while the remainder was predominantly inhibited. Despite the different proportions of excited compared to inhibited cells across cell layers, the firing rates of >85% of all recorded cells in LII and LIII were altered by the LIIP cell manipulation. Our strategy thus effectively and selectively perturbed superficial layers in the mEC.

Because the optically activated LII cells are excitatory[32,33,38,42,43], we reasoned that the profound inhibition observed across LII and LIII was mediated by synaptically connected interneurons. To test whether a substantial fraction of interneurons responded to the photostimulation, we classified cells as putative interneurons using a combination of average firing rate and waveform shape (Fig. 2f) and found that 23 of the 30 interneurons (76.7%) were excited. Next, we sought to determine whether excitation by LIIP cells was selective to any particular interneuron type. In a previous study, interneurons in the hippocampus have been sub-classified by using the burst index[44]. However, we could not find any relation between the waveform (Fig. 2f) or burst index (Fig. 2g, h) of interneurons and the type of response they demonstrated, leading us to conclude that various subtypes of interneurons were engaged by our manipulation. Consistent with the hypothesis that LIIP activation led to rapid synaptic excitatory responses across all other LII cells, followed by di-synaptic inhibition from excited interneurons, the latency for excitation was found to precede the latency for inhibition by 3–4 ms (Fig. 2i).

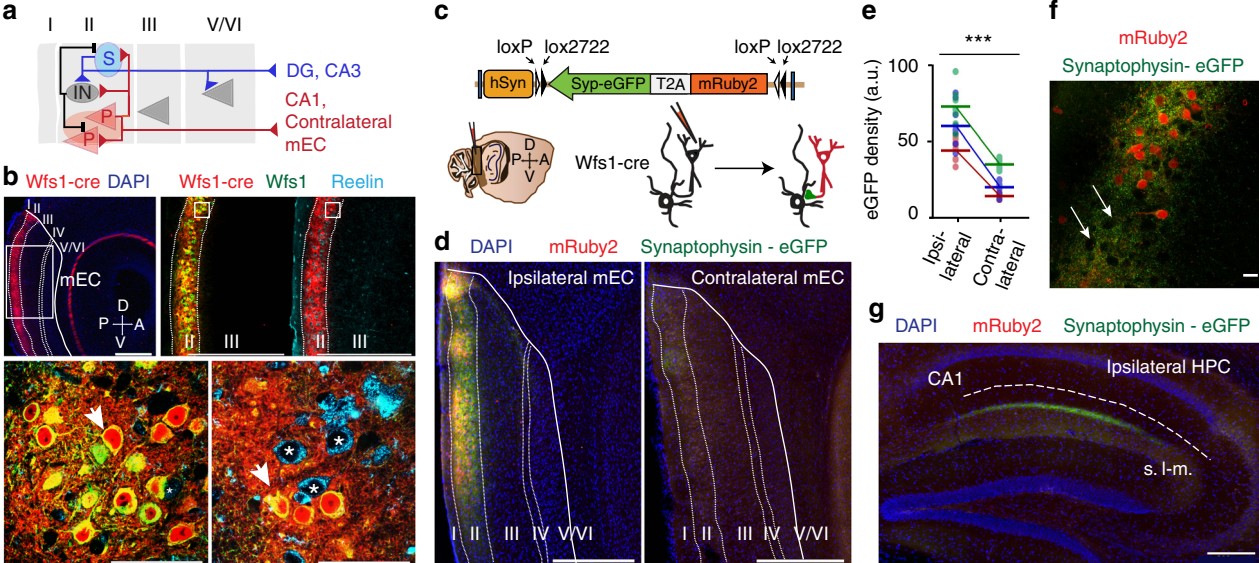

**Fig. 1** LIIP cells project densely within mEC LII. **a** Schematic describing the local anatomical connections of LII cells (P pyramidal cell, S stellate cell, IN interneuron). **b** Top, Expression of Cre recombinase in a Wfs1-creER mouse is restricted to LII of mEC, as shown by crossing the Wfs1 line to the Ai14 mouse line. Scale bar, 500 μm. Top right, Magnified images (corresponding to the white box to the left) show that Cre expression overlaps with the Wfs1 protein but not Reelin. Bottom, Magnification of Wfs1- and Reelin-stained sections (white boxes in top-right panel) shows Cre expression (red) exclusively in Wfs1+ (green) LIIP cells (yellow when the two labels are merged, arrows) but not in Reelin+ (blue) LIIS cells (asterisks). Scale bar, 20 μm. **c** An AAV construct with a Cre-dependent cassette expressed mRuby2 and Synaptophysin-eGFP, which when activated, resulted in the labeling of cell bodies with mRuby2 and of synaptic terminals with eGFP. **d** Left, Injection in the dorsal mEC of Wfs1-creER mice reveals dense synapses of Wfs1+ cells locally within LII but not extending to other layers of the mEC. Right, The contralateral hemisphere of the same brain has sparse synaptic density. Scale bar, 500 μm. **e** Quantification of eGFP fluorescence intensity reveals that the projections of LIIP cells to the contralateral hemisphere are weaker than those on the ipsilateral side [Wilcoxon matched-pairs signed-rank test, n = 27 ROIs from 3 mice, W(27) = −378, p < 0.001]. Each color represents a different animal, and individual dots represent different ROIs from each animal. ***p < 0.001. **f** LIIP cells (red) synapse densely around all cell bodies within LII. Arrows point to LII Wfs1− cells that are surrounded by LIIP cell synapses. Because LIIP cells occur in patches along LII, these Wfs1− cells are in stellate cell-rich zones between patches. Scale bar, 20 μm. **g** We observed synapses from LIIP cells in CA1 but not in the DG-CA3 region, where LIIS cells are known to terminate, further confirming that the Wfs1 line is selective for LIIP cells. Scale bar, 500 μm

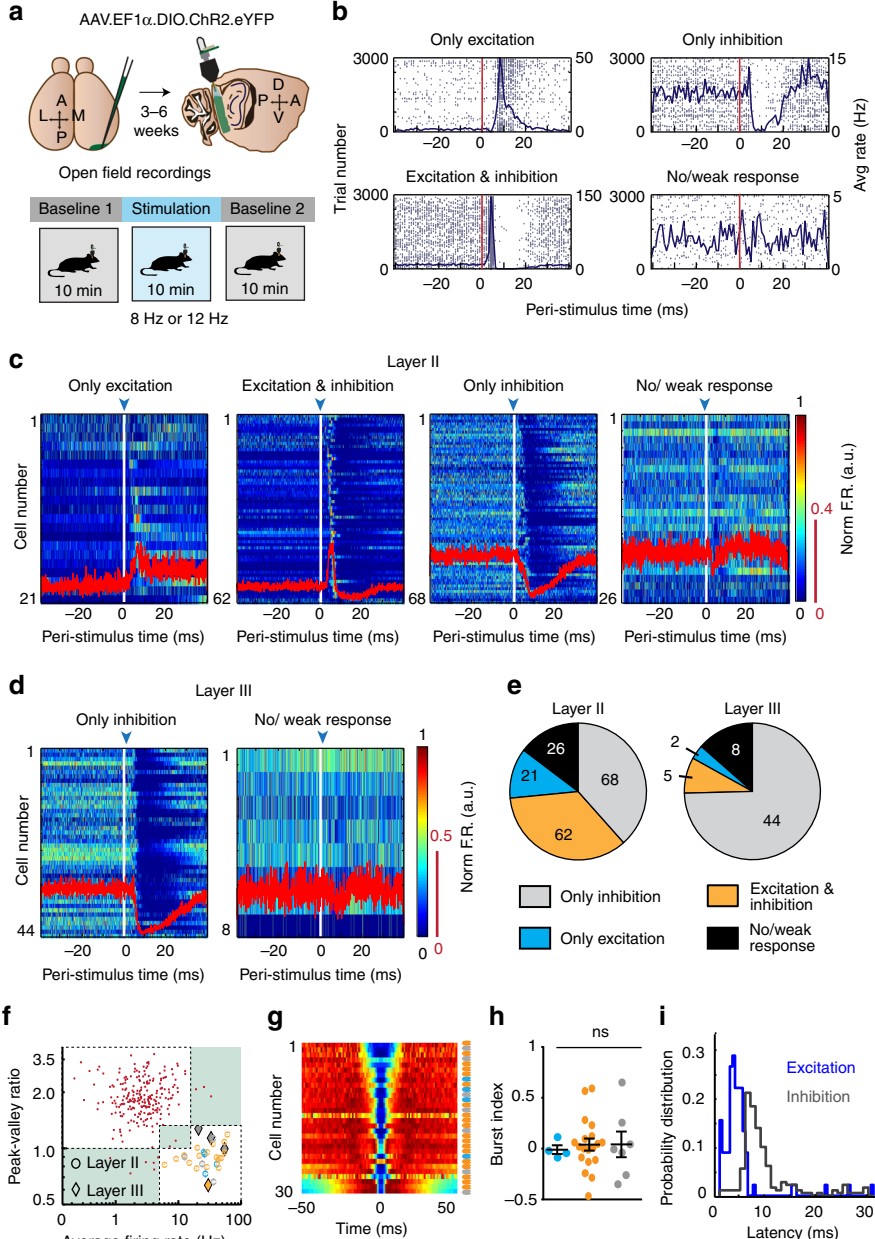

**Fig. 2** Functional connectivity of LIIP cells. **a** Schematic describing the recording strategy and behavioral paradigm. **b** Cells were divided into four categories, depending on how they responded to the photostimulation. Example raster plots and peri-stimulus time histograms for each of the categories are shown. Vertical red lines indicate light onset. **c** All recorded mEC LII principal cells divided by response category. Each row is the peri-stimulus firing rate of a single cell, normalized to its peak firing rate (scale, color bar to the right). The cells are sorted by their response latencies. Excited cells have a sharp peak within 10 ms after the light pulse, whereas inhibited cells reduce their firing rate up to ~30 ms after the light onset. Red traces, mean normalized firing rate across all cells within the category (scaled to the red bar on the right). Blue arrow, light onset. **d** Plots as described in **c** but for cells recorded in LIII. Primarily inhibition was observed in LIII cells, thus the excitatory responses of <15% of cells are not shown. **e** Quantification of responses shows that ~85% of LII and LIII cells were perturbed by the stimulation. **f** Average firing rates on a logarithmic scale versus the peak-valley ratio of the spike waveform of all cells distinguished putative principal cells (red dots) from interneurons. Cells outside of or between the two clusters were excluded from further analyses (sections in green). Thirty putative interneurons were recorded, 26 from LII (circles) and 4 from LIII (diamonds). Color code as in **e**. **g** Color-coded (maximum, red; minimum, blue) autocorrelograms of all interneurons ordered by their burst index and with the response category of each cell denoted by colored dots on the right (color scheme as in **e**). **h** The burst index (mean ± sem) of putative interneurons is plotted by response category (color scheme as in **e**). Response category did not predict burstiness [Kruskal–Wallis test, $H(3) = 3.124$, $p = 0.2097$]. **i** Distribution of the latencies for excited (blue line) and inhibited (gray line) principal cells. Excitation occurred rapidly and was followed by feedback inhibition

Since our anterograde tracing also revealed sparse synapses from LIIP cells to the contralateral mEC (Fig. 1d, e), we sought to examine the functional relevance of these projections. In a separate subset of mice, we compared the strength of ipsilateral and contralateral stimulation. The response to bilateral stimulation was almost identical to unilateral stimulation (Supplementary Fig. 3), which led us to conclude that bilateral manipulations would not perturb the ipsilateral mEC any more effectively than unilateral stimulation. Thus all further analyses were restricted to datasets with unilateral perturbation.

**LIIP cell activation transiently inhibited speed cells**. Activation of LIIP cells resulted in large-scale perturbation of cellular activity within LII and LIII. This response pattern provided us with an opportunity to address whether strong local circuit perturbation in the superficial layers interfered with the firing properties of functional cell types in the same region. We began our analysis with speed cells and compared 10–15 min sessions with repetitive light stimulation (delivered at 8 or 12 Hz for the entire session duration) to 10–15 min baseline sessions (without light stimulation) that occurred before and after the light stimulation sessions. Cells were classified as speed cells if their speed score exceeded the 95th percentile of scores from shuffled timestamps[4,45]. Using this method, 47 of the 266 recorded mEC cells across LII and LIII were identified as speed tuned, with an average firing rate of $12.8 \pm 2.36$ Hz (mean ± sem, 32 principal cells and 15 interneurons, Supplementary Fig. 4a–g). When analyzing these cells over an entire recording session with repetitive optical stimulation (at 8 or 12 Hz) of LIIP cells, speed tuning was not altered, regardless of the stimulation protocol and whether a cell was excited or inhibited by the perturbation (Fig. 3a–c).

Because our optical manipulation only had measurable effects on firing rates for ~25 ms, we next considered the possibility that speed tuning could only be transiently perturbed. For this analysis, we reasoned that spiking elicited by direct activation of ChR2 trivially alters the firing patterns of cells in the local circuit. However, during the phase when cells rebounded from the subsequent feedback inhibition, we could examine the recovery of information coding by processing of afferent and local inputs to the network. We therefore sorted all speed cells based on their

responses to light and asked whether inhibited cells lose their speed tuning during the period of maximum inhibition (I, from the minimum firing rate until each cell regained 50% of its average baseline firing rate). As a baseline, we downsampled spikes occurring immediately before light onset (Pre) to match the total number of spikes in I. The instantaneous running speeds during each of these windows were then used to perform a correlation with firing rates in the corresponding window (Fig. 3d). The resulting speed scores did not differ between I and Pre but were in both cases reduced compared to baseline, suggesting that subsampling to low firing rates is sufficient to lower the score (Fig. 3e). Indeed, the speed scores during I were correlated to the number of spikes in the window (Fig. 3f). Therefore, the decrease in speed tuning of mEC speed cells did not exceed the level that could be explained by reduced spiking during the inhibition window.

**Theta oscillations were unaltered by LIIP cell activation**. We next asked whether other potential sources of speed information within the mEC could be perturbed during our manipulation. Theta oscillations arising from the medial septal area are modulated by running speed[18] and can therefore be a source for speed information. For our analysis of theta oscillations, we had to consider that our optogenetic stimulation generated population spikes in the local field potential (LFP) signal, which were superimposed onto endogenous LFP and did not allow us to measure theta power (6–10 Hz) independently of stimulation effects for 8 Hz stimulation (Fig. 4a, b). For 12 Hz stimulation

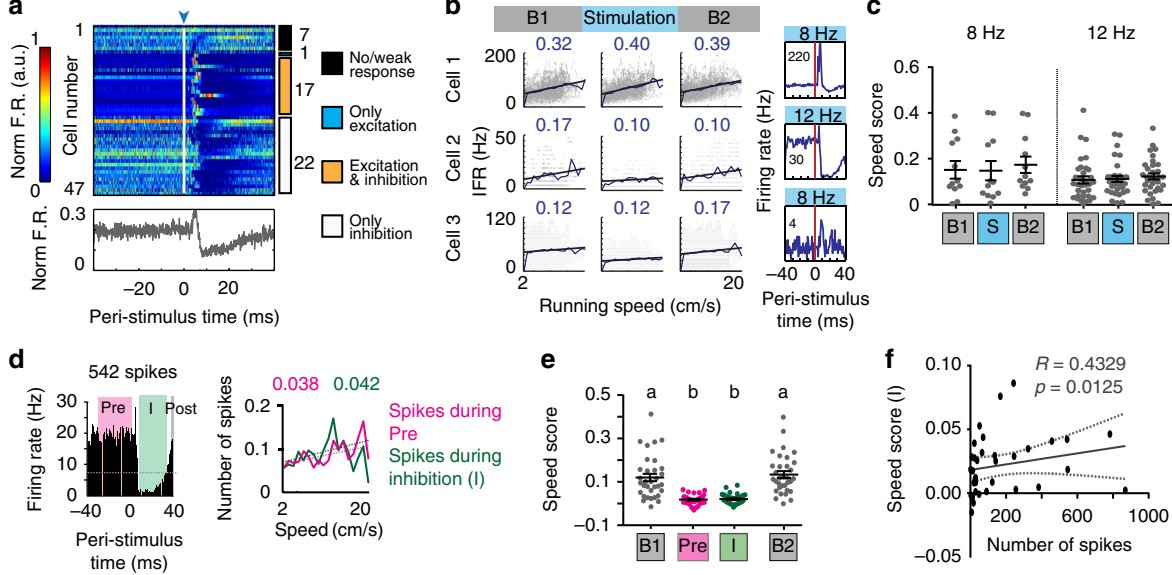

**Fig. 3** Speed cells were transiently inhibited by LIIP stimulation. **a** Peri-stimulus time plot of all speed cells (32 principal cells, 15 interneurons). Each row is a cell with the firing rate scaled as in the color bar to the left. Cells are ordered by optogenetic response category with the number of cells in each category indicated to the right. The mean normalized firing rate over all speed cells is depicted below the plot, demonstrating a period of inhibition. **b** Example speed cells retained their speed scores (number on top of each plot) during the stimulation session regardless of response category. Optogenetic stimulation frequency is provided on top and peak firing rate is printed within the PSTH plots to the right. **c** Speed scores for stimulation sessions did not differ from baseline sessions. Each dot represents a cell [$n = 12$ cells for 8 Hz sessions, 33 cells for 12 Hz sessions from 7 mice; two-way ANOVA with repeated measures on one factor, $F_{(2,86)} = 0.2367$, $p = 0.7898$]. **d** Left, PSTH of a speed cell that was transiently inhibited by the optogenetic stimulation. Right, Correlation between instantaneous running speed and either the number of spikes during the period of maximum inhibition (I, green), or an equal number of spikes randomly sampled before light onset (Pre, magenta). Speed scores are shown above the plot. **e** Quantification of speed scores revealed no difference between Pre and I, but both scores were reduced compared to baseline [one-way repeated-measures ANOVA, $n = 33$ cells from 7 mice, $F_{(1.858,59.45)} = 44.94$, $p < 0.0001$]. Significantly different groups (Tukey's multiple comparisons test, $p < 0.05$) are assigned different letters (a, b). **f** There was a significant correlation between the speed score during inhibition (I) and the number of spikes during inhibition (I), suggesting that the decrease in speed score can be explained by the increased variability from low spike numbers [Spearman's rank correlation, $n = 34$ cells from 7 mice, $R = 0.4239$, $p = 0.0125$]. Plots in **c**, **e** show the mean ± sem

sessions, however, we could clearly distinguish two separate frequency bands—one at 12 Hz, arising due to the population spikes and another at theta frequencies—and were able to show that there was no detectable increase in the endogenous theta amplitude from population spikes (Fig. 4c). We next performed a linear regression between running speed and either the peak theta frequency (Fig. 4d) or average theta power (Fig. 4e) and found no difference between the slopes of baseline and stimulation sessions for both measurements for 12 Hz sessions. For 8 Hz sessions, population spike artifacts remained in the filtered signal (6–10 Hz) and increased the average theta power, while the power nonetheless remained modulated by running speed to the same extent as during baseline. Theta-related speed information was thus retained in the mEC throughout sessions with local mEC circuit perturbation. In addition, we observed that mEC cells that were strongly modulated by theta oscillations maintained their theta phase preference during feedback inhibition (Supplementary Fig. 5).

**HD cell tuning was intact during LIIP cell activation.** We next investigated whether LIIP cell photostimulation resulted in a disruption of HD tuning. Because the HD cell population in our dataset showed a clear bimodal distribution of tuning width (Supplementary Fig. 4a–c), we classified sharply tuned cells (HD score >0.5, $n = 23$ cells across LII and LIII) as narrow HD cells and the less strikingly HD modulated population as broad HD cells (HD score more than the 95th percentile of shuffled

distributions and <0.5, $n = 58$ cells across LII and LIII). We first examined the effect of stimulation on narrow HD cells. There was no change in HD selectivity for the entire stimulation session compared to baseline sessions, regardless of stimulation frequency and response type (Fig. 5a, b). We next determined whether there was any disruption in the firing properties of these cells on a shorter time-scale. Surprisingly, we found that most narrow HD cells (15 of 23, 65.2%) only showed weak or no responses to light stimulation (Fig. 5c), while other cells recorded in proximity or simultaneously with narrow HD cells were highly responsive to light stimulation (Supplementary Fig. 6, Supplementary Table 2). Because narrow HD cells were largely non-responsive, we could not use an inhibition window as described for speed cells and instead compared spikes during the time window when other cells in the circuit were maximally inhibited (between 10 and 25 ms after light onset) to an equal number of spikes between 15 and 0 ms before light onset (Fig. 5d). The HD score was not significantly changed even during the smaller time window (Fig. 5e).

Next, we examined the effect of LIIP cell activation on broad HD cells. As for narrow HD cells, HD tuning was stable when averaging across the entire stimulation session (two-way analysis of variance (ANOVA) with repeated measures on one factor, non-significant interaction, $n = 19$ cells for 8 Hz, 33 cells for 12 Hz across 7 mice, $F(2,100) = 2.192$, $p = 0.1170$, Fig. 6a). However, broad HD cells showed strong responses in firing rate to the photostimulation with approximately 85% of cells inhibited (Fig. 6b). As for speed cells, we therefore next examined whether

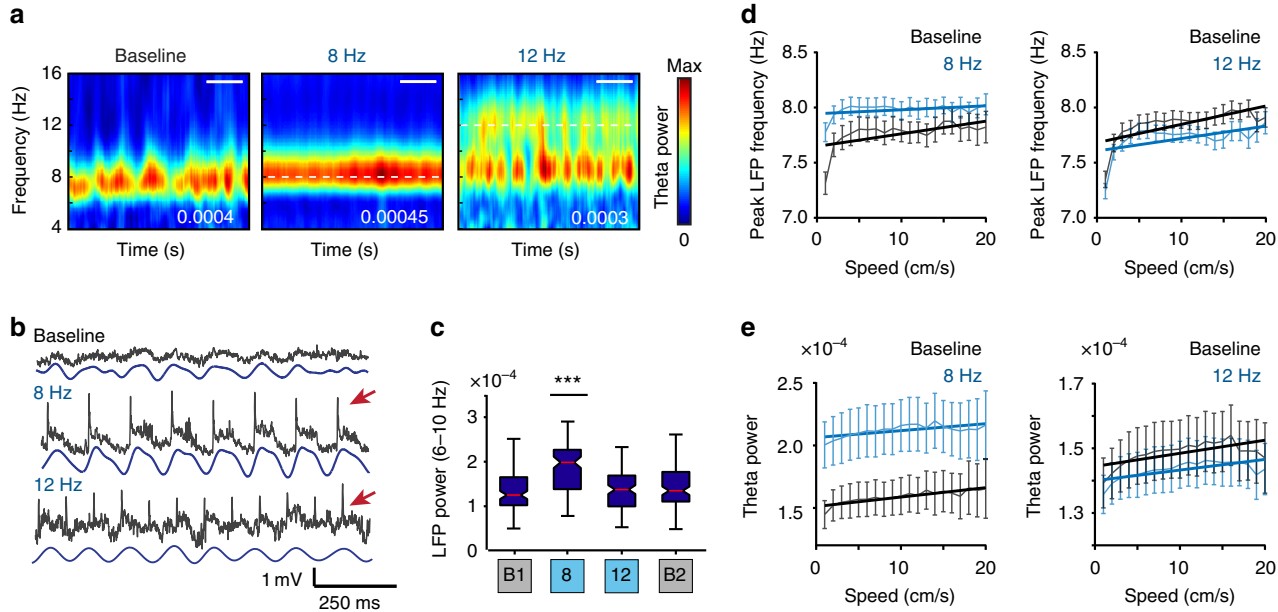

**Fig. 4** Speed modulation of theta oscillations was preserved during LIIP activation. **a** Spectrograms of wavelet power for an example baseline, 8 Hz, and 12 Hz stimulation session. The peak power is on the bottom right, and the white dashed line highlights the stimulation frequency. Scale bar, 50 s. **b** Example raw (gray) and filtered (blue) LFP traces during baseline, 8 Hz, and 12 Hz stimulation sessions, where population spikes (example shown by a red arrow) distorted the raw LFP signal during stimulation. For 12 Hz stimulation sessions, filtering the LFP between 6 and 10 Hz eliminated distortion of the signal by population spikes. **c** Average wavelet power confirms that endogenous theta power was not altered during 12 Hz stimulation. Light-induced population spikes distorted the signal during 8 Hz stimulation sessions, leading to artificially high LFP power in the theta frequency range [boxes, median ± 25th percentile, whiskers, min. to max.; one-way ANOVA, $n$ (session) = 121 (B1), 37 (8 Hz), 58 (12 Hz), and 120 (B2) across 7 mice, $F(3,335) = 7.001$, $p = 0.0001$]. Tukey's multiple comparisons test, ***$p < 0.001$. **d** Linear regression of running speed versus peak theta frequency between 6 and 10 Hz for baseline (left, gray) or 8 Hz sessions (left, blue) and baseline (right, gray) or 12 Hz sessions (right, blue). The two slopes were not different from each other in either stimulation condition [ANCOVA, 8 Hz: $n = 37$ sessions across 7 mice, $F(1,36) = 3.407$, $p = 0.0731$, 12 Hz: $n = 58$ sessions across 7 mice, $F(1,36) = 0.7171$, $p = 0.4027$]. **e** Linear regression of running speed versus mean theta power between 6 and 10 Hz for all baseline (left, gray) or 8 Hz sessions (left, blue) and baseline (right, gray) or 12 Hz sessions (right, blue). The two slopes were not different from each other in either stimulation condition [ANCOVA, 8 Hz: $n = 37$ sessions across 7 mice, $F(1,36) = 1.461$, $p = 0.2347$, 12 Hz: $n = 58$ sessions across 7 mice, $F(1,36) = 0.5918$, $p = 0.2928$]. Plots in **d**, **e** show the mean ± sem

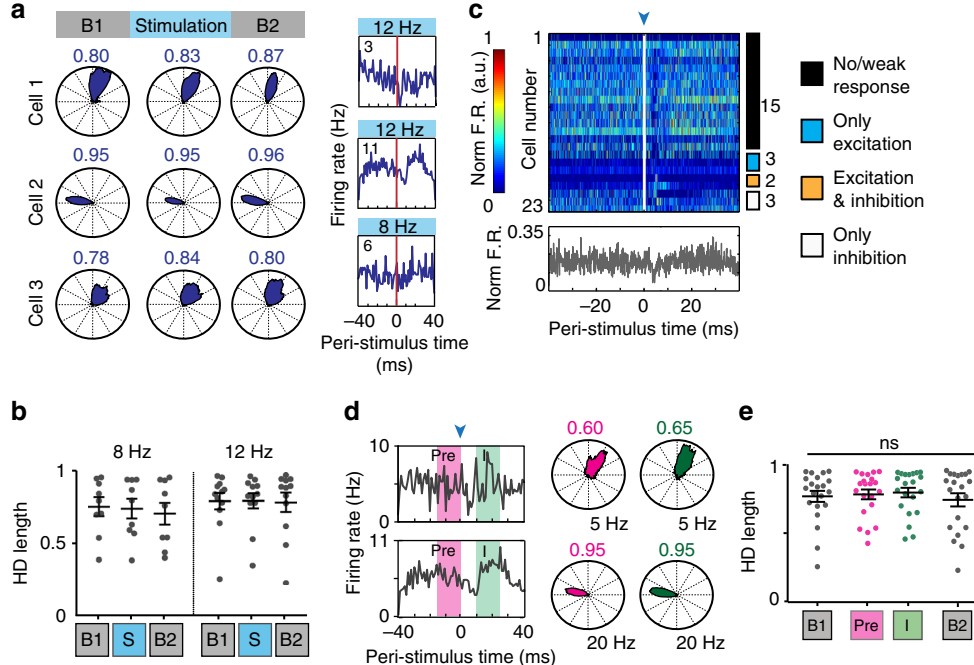

**Fig. 5** Narrow HD cells only weakly responded to optical LIIP activation. **a** Example narrow HD cells maintained their directional tuning and at most weakly responded to LIIP stimulation (shown in the rightmost plot). The HD score (i.e., mean resultant length of HD tuning) is indicated above each polar plot. **b** There was no reduction in HD score for 8 and 12 Hz stimulation sessions [two-way ANOVA with repeated measures on one factor, n.s. interaction, $n = 9$ cells for 8 Hz, 12 cells for 12 Hz across 6 mice, $F(2,38) = 0.3363$, $p = 0.7165$]. **c** Peri-stimulus time plot for all HD cells. Cells are ordered by their optogenetic response category with the number of cells in each category indicated to the right. Most narrow HD cells showed weak or no response to the optic stimulation, as can also be seen from the mean normalized firing rate of all narrow HD cells. **d** HD tuning of two example cells when only spikes between 10 and 25 ms after light onset (I, green) and the same number of spikes between 15 and 0 ms before light onset (Pre, magenta) were included. **e** Quantification of HD tuning of all narrow HD cells revealed no difference between the two baseline sessions, and I and Pre windows [one-way repeated-measures ANOVA, $n = 21$ cells from 6 mice, $F(2.199,43.97) = 2.493$, $p = 0.0894$]. Plots in **b**, **e** show the mean ± sem

HD tuning was affected during the period of maximal inhibition (I) and, for comparison, randomly selected an equal number of spikes from the 30 to 0 ms period before each light pulse (Pre) (Fig. 6c). We observed that broad HD cells were not disrupted in their HD tuning during I compared to Pre (Fig. 6d). Additionally, we found that the baseline HD tuning was higher for cells that received less inhibition (Fig. 6e). These results were also reproduced with a more conservative, 99th percentile threshold to classify broad HD cells (Supplementary Fig. 4h-j). Together with the finding that sharply tuned HD cells are largely excluded from inhibition, we thus find an inverse relation between the extent of baseline HD tuning and local feedback inhibition and that HD tuning curves were preserved irrespective of the level of inhibition.

**LIIP cell activation led to transient errors in grid firing.** Because grid cells are thought to depend not only on circuits within the mEC that convey direction and speed information but also on recurrent connections between grid cells[8–12], it is feasible that grid cells show a more major disruption compared to other cell classes. We identified 52 grid cells (grid scores >95th percentile of shuffled distribution[45]), all of which were recorded from tetrodes located in LII. We began by examining the responses of grid cells to light pulses (Fig. 7a) and found that most grid cells responded either with excitation and inhibition or with only inhibition, in similar proportions as the entire LII cell population. We first analyzed grid patterns throughout entire recording sessions with light stimulation and found no obvious disruption in grid firing, regardless of the stimulation frequency and the response type of each cell (Fig. 7b, c).

As a precise measurement of the accuracy of grid patterns during the brief periods of optogenetically induced excitation and inhibition, we calculated the fraction of spikes occurring outside of grid fields compared to the total number of spikes. As expected, only a small fraction (~20%) of grid cell spikes were outside of grid fields during baseline sessions. Next, we divided grid cells into two non-exclusive groups—cells that were excited ($n = 29$) and cells that were inhibited ($n = 39$). For the excited cells, we expected the trivial result that spikes that were elicited as a result of direct optogenetic stimulation would be randomly dispersed because the light stimulation pattern was randomly distributed across open field locations. Accordingly, we found that the out-of-field firing directly due to excitation (spikes between 0 and 10 ms after light onset) was substantially increased (Fig. 8a, b) and that there was only a small remaining degree of preference for spiking within grid fields (Supplementary Fig. 7a).

To examine information coding by local processing, we next examined spikes that occurred during the period of maximal feedback inhibition (I) in the superficial layers. When examining the spatial distribution of these spikes, we found that spike locations during this period were less accurate than those of an equal number of randomly selected spikes from the control period (Pre, 30–0 ms before light onset) (Fig. 8c, d). The accuracy of grid cell spikes rapidly returned to baseline levels as the grid cells emerged from the period of inhibition (Fig. 8e, Supplementary Fig. 7b, c). To further exclude the possibility that the increase in out-of-field firing emerged as a consequence of directly activating cells, we separately examined grid cells that were only inhibited ($n = 16$) and found that the reduction in accuracy was equally

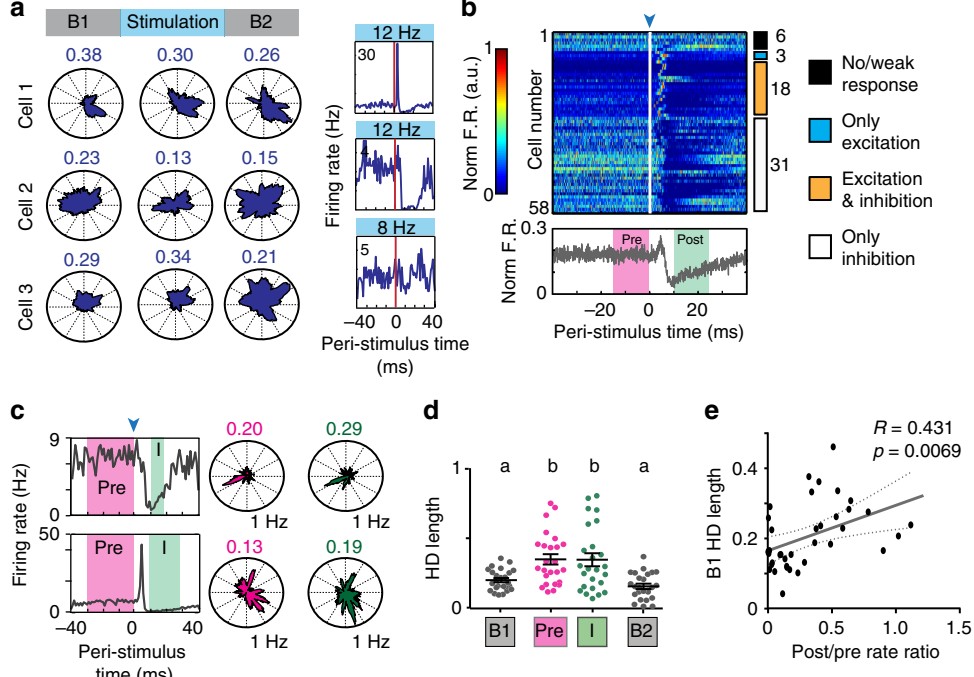

**Fig. 6** Inhibition of broad HD cells did not reduce HD tuning. **a** Example broad HD cells retained their directional tuning regardless of their response to optical LIIP activation (shown in the rightmost plot). The HD score is indicated above each polar plot. **b** Peri-stimulus time plot for all broad HD cells. Cells are ordered by optogenetic response category with the number of cells in each category indicated to the right. Broadly tuned HD cells displayed heterogeneous responses to stimulation, but a large fraction was inhibited, as also seen in the mean normalized firing rate over all broad HD cells. **c** Examples of broad HD cells that were inhibited by LIIP activation. HD scores were calculated only for spikes during the inhibition period (I, green) and for an equal number of spikes from the period before light onset (Pre, magenta). **d** Quantification of HD tuning of all inhibited broad HD cells revealed no differences between Pre and I periods, but both windows had significantly greater HD tuning than baseline [one-way repeated-measures ANOVA, $n = 25$ cells from 7 mice, $F(1.8, 43.21) = 15.46$, $p < 0.0001$], indicating that HD scores moderately increase when using fewer spikes in the analysis. Significantly different groups (Tukey's multiple comparisons test, $p < 0.05$) are assigned different letters (a, b). The mean ± sem are shown. **e** The baseline HD score of inhibited broad HD cells was correlated with the reduction in firing rate by optical LIIP cell stimulation, measured as the ratio between firing rate after light onset [10 to 25 ms, Post, green in **b**] and before light onset [−15 to 0 ms Pre, magenta in **b**]. The results imply that increased baseline HD tuning was related to lower feedback inhibition [Pearson's correlation, $n = 38$ cells from 7 mice, $R = 0.431$, $p = 0.0069$]

pronounced in these cells (Fig. 8f). We also examined conjunctive grid by HD cells ($n = 10$) and determined that grid accuracy but not HD tuning was impaired during inhibition of these cells (Supplementary Fig. 7f–j).

To ensure that our finding of transiently increased out-of-field firing was independent of field boundary definitions, we used a second metric for grid accuracy. We calculated the average distance from the closest grid node center for spikes during the window of maximal inhibition (I) and for an equal number of spikes selected from the period before each light pulse (Pre, 30–0 ms). For comparison, we also calculated the mean distance of an equal number of random locations along the path of the animal. These measures were normalized for grid field sizing as described in the Online Methods. We found that there was an increase in the average distance from the center of the node for spikes during inhibition (Fig. 9a, b, Supplementary Fig. 7e). Finally, we tested whether decreased grid accuracy was related to the amount of inhibition a cell received. The spatial displacement of spikes during I showed a weak but significant correlation with the rate reduction during I (Fig. 9c). Because more inhibited cells contribute fewer spikes to the analysis, we confirmed that the correlation did not simply arise from decreased grid accuracy when fewer spikes are available for the analysis (Supplementary Fig. 7d). Taken together, we therefore found that grid accuracy was broadly reduced during the period of feedback inhibition.

## Discussion

To test theories that grid cells arise from the processing of direction and speed information by local recurrent circuits in the mEC, we selectively perturbed neuronal activity within the superficial layers while recording from speed, HD, and grid cells. Consistent with recent results that identified direct excitatory connections between LIIP cells and other LII cells in entorhinal slices[32,33], we observed synaptic terminals of LIIP cells predominantly within the ipsilateral layer II. As expected from the dense local connectivity of LII cells, optical stimulation in freely moving mice resulted in synchronous excitation that was followed by a period of strong feedback inhibition within layer II, similar to circuit responses observed in several other cortical regions[46,47]. In addition, we also observed that the majority of layer III cells were transiently silenced by feedback inhibition. Although ~85% of mEC superficial cells were responsive to the perturbation, most sharply tuned HD cells showed no or weak responses to LIIP cell stimulation. In contrast, we found that all other cell classes, including broad HD cells, speed cells, and grid cells, were strongly excited and received feedback inhibition. During the period of inhibition, measurements of HD and speed tuning were only affected to the extent that could be predicted by the reduced spike numbers. In contrast, grid accuracy was reduced beyond the level that was expected from the lower spike numbers during the window of inhibition. However, the location errors in grid cells were brief and not cumulative, which suggests

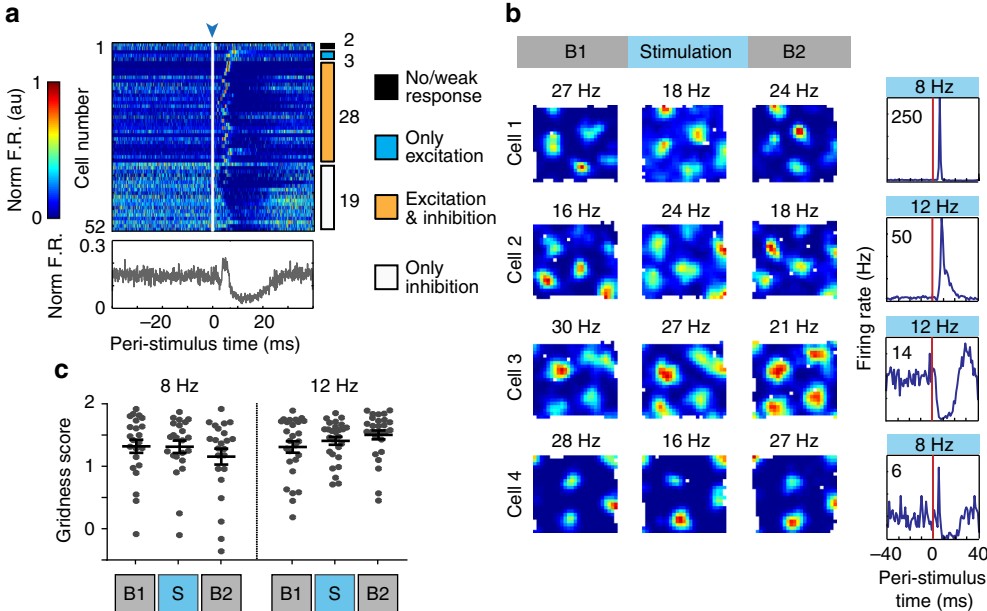

**Fig. 7** Grid cells were inhibited by LIIP cell activation. **a** Peri-stimulus time plots for all recorded grid cells. Cells are ordered by the response category to optical LIIP activation with the number of cells in each category indicated to the right. Most recorded grid cells (50 of 52) responded to the optical stimulation, and the response typically included a sustained period of inhibition (47 of the 52 grid cells). The mean normalized firing rate over all grid cells is depicted below the plot, demonstrating a sustained window of inhibition. **b** Over the entire stimulation session, example grid cells show preservation of grid fields regardless of their response to the local circuit perturbation (indicated in the leftmost plot). Rate maps are color coded from blue (minimum firing rate) to red (maximum firing rate, provided above each plot). **c** Grid scores remained high during stimulation sessions [two-way ANOVA with repeated measures on one factor, $n = 23$ cells for 8 Hz, 29 cells for 12 Hz across 3 mice, $F_{(2,96)} = 2.416$, $p = 0.0946$]. Plots show the mean ± sem

that the processing of HD, speed, and other afferent information was able to rapidly compensate for the perturbed grid tuning in the local circuit (Fig. 10). Together, our results therefore provide experimental support for theories of grid cell generation that have proposed a major contribution of local processing within the superficial layers of mEC.

It is well established that HD tuning is forwarded via the anterior thalamic nucleus and postsubiculum to mEC[15,16,48], and we find that afferent HD information is retained in an entorhinal sub-network of sharply tuned HD cells with limited functional connectivity to the majority of principal cells and interneurons. The functional separation of narrow HD cells is even more remarkable when considering that ~85% of all LII and LIII cells, including broad HD cells, responded to the LIIP cell stimulation. While our results do not identify whether narrow HD cells belong to a particular anatomical cell class, we can conclude that there is not a direct correspondence between narrow HD cells and LIIS cells. First, strong excitatory synapses have been identified between LIIP and LIIS cells[33], which predicts that LIIS cells respond to LIIP cell stimulation. Second, only ~10% of all LII cells are narrow HD cells while >50% are LIIS cells[49]. Third, non-responsive HD cells are also found in layer III. Irrespective of the anatomical identity of narrow HD cells, a local circuit that separately processes afferent HD information further could contribute to extensively integrating HD with grid and speed coding (e.g., in conjunctive grid by HD cells and broadly tuned HD by speed cells). In addition, the local segregation of specialized HD information may result in retained or even enhanced HD coding in conditions when grid firing patterns are compromised, as experimentally shown by manipulating hippocampal projections, septal projections, and local PV inputs[20,21,26].

Furthermore, the observation that speed coding was not reduced beyond the predicted level suggests that local processing is not required to further amplify or sharpen the rate-coded speed signal in mEC (see Supplementary Discussion). In contrast to the

retained speed and HD coding, our results support the notion that grid accuracy is reduced by processing of information within local recurrent networks. Recurrent network models of grid cells have either emphasized direct excitatory connectivity between principal cells or indirect inhibitory connectivity between LIIS cells[8,9,11,31,35], and both connectivity patterns have now been confirmed by pair-wise intracellular recordings from principal cells in the mEC superficial layers[31–33]. Our manipulation established that cells with both of these connectivity features (i.e., excited and inhibited, only inhibited) were perturbed by LIIP stimulation and that grid networks were not accurate during periods when inputs from other cells in the superficial layers were limited. Processing of intact afferent pathways to the superficial layers during these periods is therefore not sufficient for sustaining grid patterns, which suggests that afferent HD and speed information either needs to be further processed locally or that grid-to-grid connectivity is essential. If further processing of incoming information to the grid network were to completely explain our results, we would expect that all grid cells would be affected to a similar extent. However, we observed that effects on grid accuracy were most pronounced for grid cells that were most profoundly inhibited, which suggests that synaptic integration at either the level of the grid network or the level of individual grid cells was essential to selectively generate spikes in an accurate grid pattern.

Despite the profound perturbation of neuronal activity in most cells in the superficial layers and the inaccuracy in grid firing during the period of inhibition, we observed that grid location errors were not cumulative but were rather rapidly corrected. The lack of profoundly disrupted grid firing on a longer timescale corresponds to findings from other manipulations of local mEC circuits, which did also not observe that functional mEC cell types are disrupted with analysis over time periods of minutes[24,25]. We propose that effects on grid accuracy are short-lived because of three possibilities. First, under our recording conditions, afferent

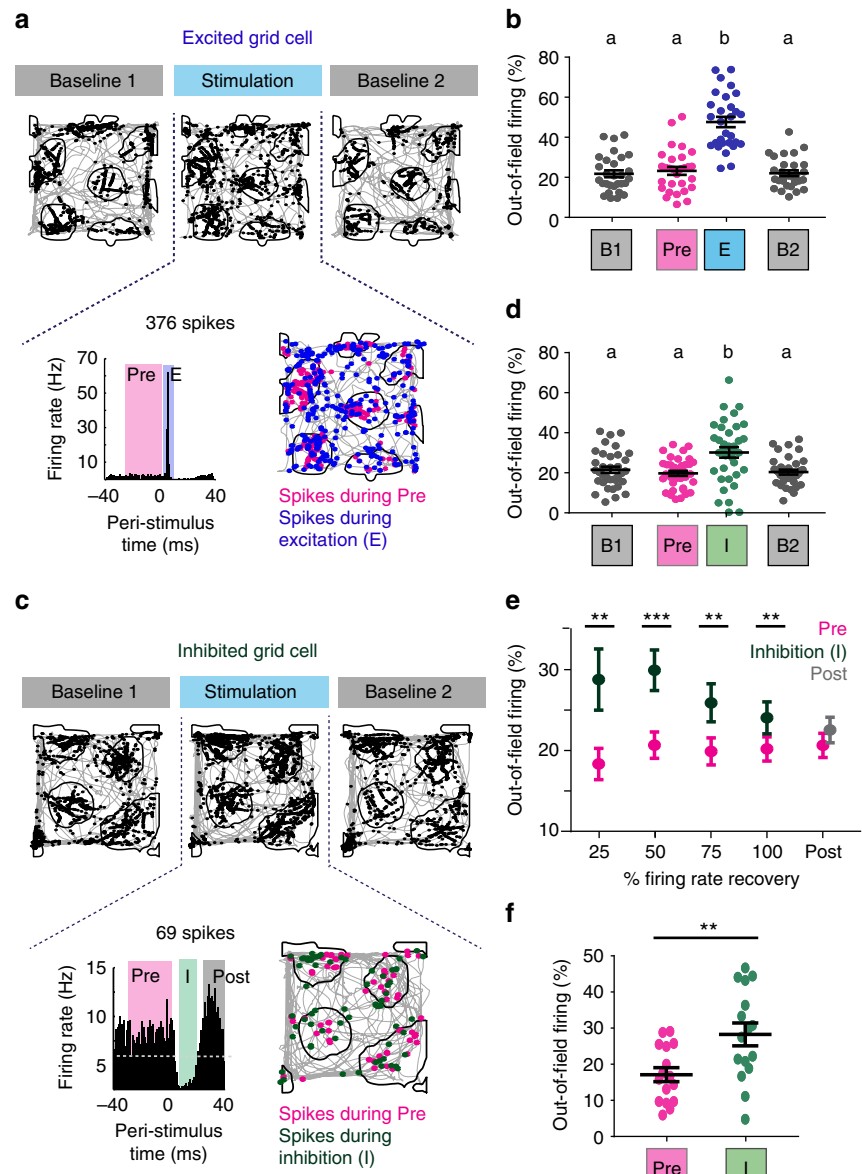

**Fig. 8** Transient inhibition of mEC superficial layers decreased grid accuracy. **a** Example grid cell excited by the stimulation. The 376 spikes firing within 10 ms after light onset (E, blue in PSTH) were dispersed in space, while an equal number of randomly selected spikes from before light onset were predominantly located within grid fields (Pre, magenta in PSTH). **b** Increased out-of-field firing of all excited grid cells confirmed the reduced accuracy of ChR2 evoked spikes [one-way repeated-measures ANOVA, $n = 29$ cells from 3 mice, $F(2.344, 65.62) = 51.84$, $p < 0.0001$]. Significantly different groups (Tukey's multiple comparisons test, $p < 0.05$) are assigned different letters (a, b). **c** Example grid cell inhibited by the stimulation. The 69 spikes during inhibition (I, green in PSTH) were less accurate than an equal number of randomly selected spikes from before light onset (Pre, magenta in PSTH). **d** Increased out-of-field firing during maximal feedback inhibition [one-way repeated-measures ANOVA, $n = 36$ cells from 3 mice, $F(1.721, 60.23) = 10.03$, $p = 0.0003$] indicates reduced grid accuracy. Significantly different groups (Tukey's multiple comparisons test, $p < 0.05$) are assigned different letters (a, b). **e** The percentage of spikes outside of grid field boundaries during different inhibition windows (up to 25, 50, 75, and 100% of recovery to baseline rate; I, green) compared to an equal number of spikes from Pre (magenta). Grid firing was restored after cells recovered from inhibition [Post, gray in **f**]. Wilcoxon matched-pairs signed-rank tests, 25%: $W(32) = -260$, $p = 0.0096$, 50%: $W(39) = -497$, $p = 0.0003$, 75%: $W(42) = -479$, $p = 0.0014$, 100%: $W(44) = -456$, $p = 0.0052$, Post: $W(44) = -150$, $p = 0.3883$. **f** Grid cells that were only inhibited and not excited also demonstrated reduced accuracy during inhibition, suggesting that reduced accuracy was not a direct consequence of ChR2 activation [paired samples $t$ test, $n = 16$ cells from 3 mice, $t(15) = 3.216$, $p = 0.0058$]. All summary plots show the mean ± sem. $^{**}p < 0.01$, $^{***}p < 0.001$

information could rapidly reset errors introduced into the firing patterns of grid cells. For example, grid cells are known to be sensitive to changes in visual cues, box shape, or box size[1,45,50,51], and it is therefore likely that a stable stream of afferent information from sensory areas, hippocampus, the contralateral mEC, or possibly even HD cells and theta oscillations, which were unaffected by our manipulation, could account for the rapid

recovery. A second possibility is that the recovery of recurrent inputs within the local mEC network to baseline firing is sufficient to enable a complete correction in grid cell firing. However, this would nonetheless require that the minor neuronal activity during the period of maximal inhibition retains sufficient information such that an updated stable attractor state can be reinstated in the mEC within about 10 ms. Lastly, it is possible that

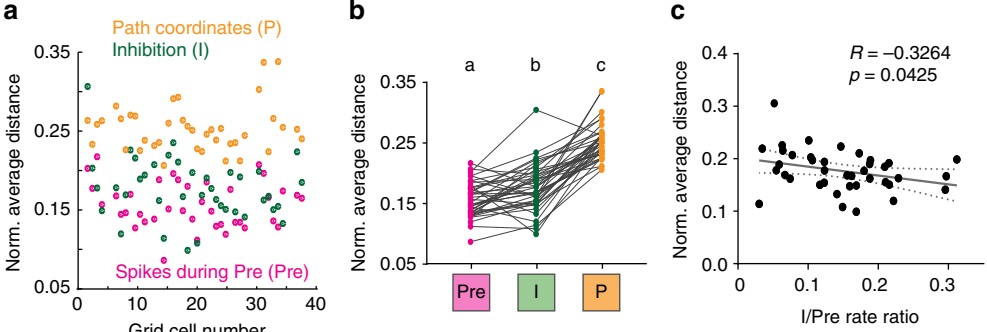

**Fig. 9** Grid cell spikes during inhibition occurred farther from the grid center. **a** The average distance of each I spike (green dots) from the center of the closest grid node was normalized to the grid spacing for each cell. This distance was compared to the normalized distance of an equal number of Pre spikes (magenta) and an equal number of random locations from the path of the animal (yellow). **b** Spikes during the period of maximal inhibition were farther away from the center of grid fields compared to control spikes but were closer than random locations along the path [one-way repeated-measures ANOVA, $n = 39$ cells from 3 mice, $F(1.707, 64.86) = 137$, $p < 0.0001$]. Significantly different groups (Tukey's multiple comparisons test, $p < 0.05$) are assigned different letters (a, b, c). **c** The normalized distance of I spikes was negatively correlated to the change in the average firing rate between I and Pre, suggesting that errors were larger for cells that were more inhibited [Spearman's rank correlation, $n = 39$ cells from 3 mice, $R = -0.3264$, $p = 0.0425$]

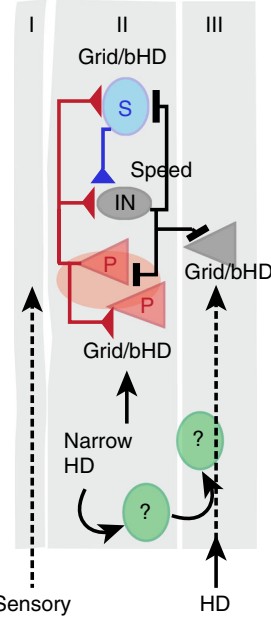

**Fig. 10** Responses of different functional mEC cell types to LII perturbation. Previous studies using slice electrophysiology have demonstrated that LIIP cells project locally within LII mEC to other LIIP cells, LIIS cells, and local interneurons (also see Fig. 1). In this study, we determined whether these extensive local projections target all functional cell types. Most cell types, including speed cells, grid cells, and broadly tuned HD cells (bHD), were found to show robust excitatory and inhibitory responses to LIIP cell activation. To the contrary, narrowly tuned HD cells only displayed weak responses to the stimulation, indicating that these cells either do not receive local inputs or are coupled to a separate local network within the mEC. Consistent with the strong afferent HD signals to the grid network and the unperturbed local HD signal in narrowly tuned HD cells, HD coding was unaffected by the perturbation, including in broadly tuned and conjunctively tuned HD cells. Similarly, speed coding was only reduced to the extent that was predicted from the firing rate reduction, including in interneurons which showed the highest degree of speed tuning. Despite the minor effects on HD and speed tuning, grid accuracy was reduced during the period of layer LII/III inhibition. Our results thus provide experimental support for the central tenet of most models of grid cell generation, which is that the key computations are performed in local mEC networks that include recurrent connections between grid cells

the error correction signal arises from grid modules much further (>1 mm) away from the few modules that we were recording and perturbing. However, this seems unlikely since grid cells across modules have been shown to remap independently of each other[51,52], which suggests that putative attractor states are distinct for each grid module and do not strongly interact with each other.

Taken together, we find that narrow HD signals form a separate subcircuit within the superficial layers, which points to a network mechanism for separate local processing of afferent HD signals that could explain the persistence of HD signals during manipulations that alter grid firing[20,21,26]. Furthermore, our results suggest that computations within recurrent circuits in the superficial layers are critical for determining the precise firing location of grid cells. While processing of afferent information by recurrent circuits between grid cells have been the standard tenet of computational models for grid firing[13,14], our results are a critical experimental confirmation that key computations for grid accuracy are performed within the superficial layers. Finally, our results provide evidence that, under circumstances of foraging with abundant sensory cues, any local perturbations of position accuracy are corrected on a timescale in the order of milliseconds.

## Methods

**Approvals**. All animal experiments were approved by the University of California, San Diego Institutional Animal Care and Use Committee and conducted according to National Institutes of Health guidelines.

**Animals**. Wfs1-creERT2 mice were obtained from The Jackson Laboratory [B6.Cg-Tg(Wfs1-cre/ERT2)2Aibs/J, Stock No.:009614]. The expression of Cre recombinase was confirmed by breeding male Wfs1-creERT2 mice with female Ai14 reporter mice, also from The Jackson Laboratory [B6.Cg-Gt(ROSA)26Sor^tm14(CAG-tdTomato)Hze/J, Stock No.:007914], and checking the expression of at least five mice across different breeding pairs. For viral tracing procedures, we used seven heterozygous Wfs1-creERT2 (four male, three female) mice. We observed no differences in the projection patterns of LIIP cells between males and females. For electrophysiology, seven heterozygous Wfs1-creERT2 male mice weighing between 25 and 32 g were used for the unilateral stimulation experiments. An additional six heterozygous male Wfs1cre-ERT2 mice were used for the bilateral stimulation experiments. All mice were between 4 and 8 months of age. To obtain Cre expression, mice were given three intraperitoneal injections of tamoxifen (0.25 mg/g body weight, Sigma-Aldrich, T5648) over 5 days. The tamoxifen was prepared as 20 mg/ml in corn oil (Sigma-Aldrich, C8267) and stored at −20 °C. The mice were housed on an inverse 12 h light/dark cycle (lights off at 8 a.m.), and all recordings were done during the dark phase. Mice were provided food and water ad libitum. They were housed in groups (maximum five per cage) before surgery but were singly housed after the microdrive implant.

**Tissue processing and immunohistochemistry**. Anesthetized mice were perfused with cold saline solution (0.9%) followed by cold pre-made 4% paraformaldehyde in phosphate-buffered saline (PBS) solution (Affymetrix USB). Brains were post-fixed for 24 h in 4% paraformaldehyde at 4 °C and then transferred to PBS solution and subsequently sagitally sectioned (40–50 μm) on a vibrating blade microtome (Leica, VT1000 S). The sections were either mounted directly on electrostatic slides or transferred to 1× PBS for immunohistochemistry. Slides were coverslipped with DAPI Fluoromount-G (Southern Biotech, 0100-20) and imaged using a virtual slide microscope (Olympus, VS120). Sections from animals that contained tetrode tracks from the electrophysiology were then un-coverslipped, stained with cresyl violet, and coverslipped with Permount (Fisher Scientific, SP15500) for better visualization of the tracks.

For sections used for immunohistochemistry, prior to the addition of primary antibodies, the slices were blocked for 1 h at room temperature (RT) with a blocking solution comprising 10% horse serum (HS), 0.2% bovine serum albumin (BSA), and 0.05% Triton-X in 1x PBS. Slices were incubated overnight at 4 °C with primary antibodies dissolved in a carrier solution made with 1% HS, 0.2% BSA, and 0.05% Triton-X in 1× PBS. After overnight incubation and 4 washes for 10 min each with 1× PBS, the slices were incubated for at least 2 h at RT with secondary antibodies in carrier solution. Next, the slices were washed four times with 1× PBS, mounted on glass slides, and coverslipped with DAPI Fluoromount-G. The following primary antibodies were used: mouse anti-reelin (Millipore Cat# MAB5364, RRID:AB_2179313, 1:1000), mouse anti-reelin (MBL International Cat# D223-3, RRID:AB_843523), and rabbit anti-Wfs1 (Proteintech Group Cat# 11558-1-AP, RRID: AB_2216046, 1:1000). All secondary antibodies (AlexaFluor488 anti-mouse, AlexaFluor405 anti-rabbit, AlexaFluor488 anti-rabbit) were obtained from ThermoFisher Scientific. Sections were imaged using a virtual slide microscope (Olympus, VS120), and higher magnification images were obtained from a confocal microscope using ×10, ×20, or ×60 magnification lenses (Olympus FluoView, FV1000).

**Fluorescence quantification**. Sections containing the ipsilateral and contralateral mEC of mice injected with AAV-hSyn-FLEX-mRuby2-T2A-Synaptophysin-eGFP virus were imaged using a virtual slide microscope (Olympus, VS120) at ×10 magnification. For accurate comparison, the two hemispheres of each animal ($n = 3$) were imaged with two laser wavelengths—532 nm (red) and 495 nm (green)—using the same laser power and exposure time for each hemisphere. Three ipsilateral sections along the medial–lateral extent of the mEC, and three corresponding sections on the contralateral hemisphere were chosen for each animal and subsequently analyzed using Fiji ImageJ[53]. To ensure overexpression of synaptophysin-eGFP within cell bodies on the ipsilateral hemisphere does not artificially elevate the eGFP signal, the red channel (corresponding to cell bodies and axons) was subtracted from the green channel for sections from both hemispheres. These subtracted regions were excluded from further analysis such that only eGFP expressed at synaptic terminals would be used for fluorescence quantification. Three non-overlapping regions of interest (ROIs) within LII were targeted to regions with maximal GFP expression per section, and the fluorescence density (GFP signal/area of ROI) was calculated for each ROI. The average eGFP density for each hemisphere per animal was obtained by averaging the GFP densities from the three ROIs across three sections (total nine ROIs). Additionally, the spread of eGFP was calculated by applying the same threshold to all ipsilateral and contralateral sections per animal and determining the average area with GFP signal exceeding the threshold.

**Viral vector injections**. All mice were anesthetized with isoflurane (1.5–2.5% in O₂) and mounted in a stereotaxic frame (David Kopf Instruments, Model 1900). Once unresponsive to toe pinch, an incision was made, and the skull was leveled to bring bregma and lambda to the same plane. A unilateral craniotomy was made over the mEC (+0.2 mm AP from the transverse sinus, +3.75 mm ML from lambda) and a glass pipette containing the virus was slowly lowered into the brain at a 6° angle (toward anterior). For anterograde tracing, AAV-hSyn-FLEX-mRuby2-T2A-Synaptophysin-eGFP virus (provided by Dr. Byung Kook Lim, titer: ~1.5 × 10¹⁵) was injected at two locations (−1.8 mm DV and −0.9 mm DV from the surface of the brain) with 125 nl injected at each site. The pipette was left in place for 5 min at the most ventral location and then brought up to the second injection site for another injection. The injection was administered at a flow rate of 100 nl/min using a microsyringe pump (World Precision Instruments, UMP3 UltraMicroPump). For optogenetic experiments, the same injection strategy was used but with three 200 nl infusions of AAV-EF1α-DIO-hChR2(H134R)-eYFP (provided by Dr. Byung Kook Lim, titer: ~1.6 × 10¹⁴) at each site (−1.8, 1.3, −0.9 mm DV), either unilaterally or bilaterally. All viruses were packaged into AAV2/DJ serotypes. The constructs were allowed to express for at least 3 weeks before either a perfusion or a subsequent surgery to implant the mouse with a microdrive.

**Microdrive building and surgery**. For in vivo electrophysiological recordings, we used a microdrive with four tetrodes, each made of four 17 μm platinum–iridium (90/10%) wires (California Fine Wire Company). The microdrive was fitted with an optic fiber attached to the tetrode bundle (optetrode[40]). We assembled a custom

optic fiber using a 200 μm diameter optic fiber (Thor Labs, 0.50 NA multimode fiber) glued to a zirconia ferrule (Precision Fiber Products, 230 μm). The tetrodes were cut to end <1 mm deeper than the optic fiber tip in order to ensure light activation of all cells surrounding the recording site. Prior to the surgery, the tetrodes were plated with platinum to lower impedances to 125–325 kΩ at 1 kHz. Mice were prepared for surgery as described above. In order to firmly fix the microdrive and dental cement to the skull, five anchor screws were placed on the skull. A craniotomy was prepared at the implantation coordinates (+0.2 mm AP from the transverse sinus, +3.75 mm ML from lambda), and the dura was removed. Tetrodes were placed as close to the transverse sinus as possible and gently lowered ~700 μm into the brain, with the drive positioned at a 6° angle (toward anterior). In mice with bilateral stimulations, an optic fiber cannula was implanted at a depth of 0.8 mm and at a 6° angle (toward anterior) over the contralateral mEC. The drive was then secured to the skull with dental cement, and the mice were given buprenorphine (0.02 μg/g body weight, subcutaneous) for pain relief. The mice were allowed to recover for 7 days before the start of recordings, and postoperative care was administered every day until the mice were perfused.

**Recording procedures and light delivery**. Mice implanted with an optetrode were trained to forage freely in an open field (75 × 75 cm², black walls with a prominent white cue card) placed in a room with dim lights and several distal cues. Once the mice were familiar with the recording set-up, each mouse was run for at least three sessions per day—one baseline, followed by an 8 or 12 Hz stimulation session, followed by a second baseline session. Each session was approximately 10 min long but could be up to 15 min long, depending on whether the entire box had good coverage by the mouse. Recordings were performed only when well-isolated cells could be observed. Depending on the stability of cells from the previous day, tetrodes were advanced in 50 μm steps at the end of the day to ensure that the same cells were not recorded for >2 days in a row. Tetrodes were generally advanced until all tetrodes crossed layer I (characterized by a loss of signal and an increase in noise), or until the end of the drive was reached (~3 mm). During the recordings, the microdrive was connected through a multichannel, head-mounted preamplifier to a digital Neuralynx recording system (Neuralynx, Bozeman, MT). A pulley system was designed to counteract the weight of the tether and headstage. Unit activity was amplified (5000–20,000×) and band-pass filtered (0.6–6 kHz) to isolate spikes. Spike waveforms above 35–45 μV were time-stamped and digitized at 32 kHz for 1 ms. Continuous LFP was recorded from each tetrode by band-pass filtering (0.1–900 Hz) and sampling at 32 kHz. Position data of a red and a green LED located on either side of the head-mounted preamplifier were tracked at 30 Hz by a video camera mounted above the experimental area to determine the mouse's x–y position and HD.

During optical stimulation, light was delivered to the mEC using a 473 nm wavelength Blue DPSS Laser System through a mono fiber optic patchcord (Doric Lenses, MFP_200/240/1100-0.22_10m_FC-ZF1.25(F), 200 μm core, 0.22NA) or a branched fiber optic patchcord (BFP(2)_200/240/900-0.22_10m_FCM-2xZF1.25 (F)). The output intensity was adjusted to elicit an evident response in the unit recordings while minimizing large population spikes (power was usually maintained at <1 mW at the end of the patchcord). Light was delivered at 8 Hz (either 62.5 ms ON, 62.5 ms OFF or 10 ms ON, 115 ms OFF) or 12 Hz (either 41.7 ms ON, 41.7 ms OFF or 10 ms ON, 73.3 ms OFF) using the Neuralynx Spectralynx system. We found that altering the light ON duration had no effect on the response and therefore combined all stimulations performed at a certain frequency, regardless of the duration of the light.

**Spike sorting and cluster quality**. We used a customized version of the spike sorting software MClust, MATLAB 2009b (Redish, A.D. MClust. http://redishlab.neuroscience.umn.edu/MClust/MClust.html). Clustering was performed manually in two-dimensional projections of the parameter space using waveform amplitude, waveform energy, and the peak-to-valley difference. The same cell was tracked across all sessions within a day but was not tracked across days (<10% of all cells were recorded for >1 day). Cluster quality was assessed by calculating the L-ratio and isolation distance (Supplementary Fig. 1) for each cluster across baseline and stimulation sessions[54].

**Anatomical localization of cells**. Based on tetrode tracks observed in cresyl violet sections, recordings for each day were localized to a certain layer of the mEC (Supplementary Fig. 1). Because tetrodes in our microdrives were usually only ~100 μm apart and moved parallel to the cortical surface, precise localization to an entorhinal layer was possible in most animals. In the animals where the tetrodes crossed between layers, we localized cells to either layer II or III based on records of the advancement of tetrodes through the brain. Electrode localization was further confirmed by the presence of population spikes in the LFP during each stimulation session, which were only prominent when the tetrodes were in LII but not in other layers.

**Classification of cells excited by optical stimulation**. Timestamps of the TTL signal for the onset of each light pulse were used to generate peri-stimulus time rasters (PSTR) and peri-stimulus time histograms (PSTH) for each cell with a time window of ±40 ms and either 0.1 ms or 1 ms time bins. All further analysis of cell

responses was performed using either the PSTR or PSTH. Because individual cells could be both excited and inhibited, the two classifications were performed independent of each other (Supplementary Fig. 2).

The average firing rate within the 40 ms time window immediately before each light pulse (Mean$_{pre}$) was first calculated. Next, the firing rate in each 1 ms bin before the light pulse (−40 to 0 ms) was normalized to Mean$_{pre}$, and the bin with the maximum was selected. A cumulative distribution of these maximum baseline values over all cells was then generated. A cell was classified as excited if the normalized peak rate in any bin after the light onset (0–40 ms) exceeded the 95th percentile of the baseline maxima. The 95th percentile corresponded to a 1.58-fold increase (Supplementary Fig. 2a, b). Once cells were classified as excited, we examined whether it was feasible to distinguish between direct ChR2 activation and indirect activation via synaptic connections with ChR2-excited cells. A parameter commonly used to classify cells as directly excited is their reliability to spike after every light pulse. However, we found that the probability of spiking was strongly correlated with the baseline firing rate of a cell (Supplementary Fig. 2d, e). Because of the dependence of spiking probability on an inherent firing property of the cell, we did not consider a biased parameter useful for classification. We next asked whether it was feasible to use spike latency or spike jitter (i.e., the variability of latencies across trials) with the expectation that directly excited cells are characterized by low latencies and low jitter. The latency was defined as the first 0.1 ms time bin when the increase in the firing rate exceeded half of the peak rate after light onset. For calculation of the jitter, the latency for the cell to first spike was determined ($L_i$) for each light pulse. If the cell did not spike within 10 ms, the trial was assigned NaN (not a number). The jitter was defined as the standard deviation of the latencies across all trials (Supplementary Fig. 2c). We found that almost all excited LII cells responded at extremely short latencies and could not be separated into categories based on the latency. Similarly, while the jitter was more variable in our dataset, there was no obvious cutoff (Supplementary Fig. 2g). The ambiguity between direct and synaptic excitation is consistent with a recent study that performed paired recordings between LIIP and LIIS cells in entorhinal slices and demonstrated that the latency of synaptic excitation between LIIP and LIIS cells can be as low as 1 ms[33]. Therefore, we considered that cells cannot be classified as directly versus indirectly excited, consistent with reports that direct excitation and synaptic excitation are not readily distinguishable in circuits with recurrent connectivity[55,56]. However, we used direct and indirect excitation of cells in the mEC superficial layers as an opportunity to elicit feedback inhibition to perturb local circuits, similar to the rationale that has been used for electrical stimulation of recurrent collaterals[57–60].

**Classification of cells inhibited by optical stimulation.** The baseline firing rate (Mean$_{pre}$) was calculated for the 40 ms window before the light pulse. Next, a 10 ms wide sliding window was advanced with a step size of 1 ms over the 40 ms before light onset, the average within each window was normalized to Mean$_{pre}$, and the window with the minimum normalized rate was selected. A cumulative distribution of these baseline values was then generated over all cells. A cell was classified as inhibited if the mean normalized rate in any 10 ms sliding window after light onset (0–40 ms) was less than the 95th percentile of the baseline minima. The 95th percentile corresponded to a decrease to 66% of Mean$_{pre}$ (Supplementary Fig. 2h, i). The latency of inhibition was defined as the first bin of the 10 ms window with the minimum firing rate. As expected, the latency for excitation across all cells preceded the latency for inhibition (Fig. 2i).

**Interneuron analysis.** Interneurons were distinguished from principal cells by using a minimum average firing rate cutoff of 5 Hz and a maximum peak-valley ratio of 1.3 (Fig. 2f). This criterion therefore ensured that putative interneurons had a symmetrical waveform shape with approximately matching peak and valley amplitudes, as would be expected for inhibitory cells. Cells that could not be unambiguously classified were excluded from all further analyses. Interneurons have previously been divided into putative subtypes by their spike-burst magnitude[44], and we implemented the same analysis. Briefly, the burst index of each cell was defined based on the spike time autocorrelogram and these parameters were used to sort interneurons.

**Rate maps.** The recording box was divided into $3 \times 3$ cm$^2$ bins. Firing rate maps were constructed by dividing the total number of spikes in each bin by the total time the animal spent in the same bin. Rate maps were smoothed with a $5 \times 5$ pixel filter [0.0025 0.0125 0.0200 0.0125 0.0025; 0.0125 0.0625 0.1000 0.0625 0.0125; 0.0200 0.1000 0.1600 0.1000 0.0200; 0.0125 0.0625 0.1000 0.0625 0.0125; 0.0025 0.0125 0.0200 0.0125 0.0025], and bins that were >2.5 cm away from the tracked path or had a total occupancy of <150 ms were assigned NaNs in the rate map.

**Firing field boundaries.** For each rate map, field boundaries were defined on the reference map (average map from two baseline sessions and a stimulation session) by building contours from the bin with the maximum firing rate outwards until a threshold value of 0.3 times the maximum was reached. If any peaks >2 Hz remained, the procedure was repeated. For cells with multiple fields, contours were then recalculated simultaneously for all fields, such that the edge of each field was

defined as the contour at which the threshold value was reached or where two fields met, whichever came first.

**Grid score and spatial autocorrelation.** Grid cells were identified by calculating a gridness score (i.e., six-fold symmetry in a cell's spatial autocorrelation)[22]. First, rate maps were generated with $1.5 \times 1.5$ cm$^2$ size bins and used to generate a spatial autocorrelation matrix. An annulus that contained the first hexagon of peaks around the center but excluded the central peak was then retained in the matrix, and a second correlation was performed by rotating the annulus to different degrees with respect to itself. If there is six-fold symmetry, correlations at relative phase offsets of 30, 90, and 150° would be low, whereas correlations at 60, 120 and 180° would be high. The gridness score was calculated as the difference between the average of the first and the average of the second set of values. Finally, we tested whether the score exceeded values from shuffled data[45].

**HD score.** The HD of the animal was calculated from the angle between the two diodes in the horizontal plane[22]. The number of spikes within each 1° angular bin was divided by the time spent at that angle to generate polar plots. The mean vector length of these plots is the HD score.

**Speed score.** Speed cells were classified based on speed scores[4], which were obtained by calculating the Pearson product–moment correlation between the cells' firing rate and the instantaneous running speed in 33 ms bins (corresponding to the acquisition rate of position data). Firing rates were smoothed with a 250 ms wide Gaussian filter, and instantaneous running speed was obtained using a Kalman filter on the position data[4].

**Cell-type identification.** Classification of a cell as a grid, HD or speed cell was performed by determining whether the respective score was greater than the 95th percentile of the cell's scores from shuffled data[45]. Shuffled spike trains for each cell were generated by randomly selecting a value between 20 and 580 s and by then adding the value to the timestamp of each spike. Spike times exceeding the total duration of a session were wrapped to the beginning of each session. Shuffling was repeated 1000 times for each cell in each session. In addition, HD cells were further sub-classified into narrow and broad HD cells based on the finding that there was a clear cluster of narrowly tuned cells with scores >0.5 (Supplementary Fig. 4). Because each cell was recorded for at least three sessions (stimulation as well as baseline before and after stimulation), we had to account for the possibility that a cell was not consistently classified across sessions. To assign a cell to a particular functional cell type, we therefore required that the respective score was above the 95th percentile cutoff in at least two out of the three sessions.

**Measuring effects of optical stimulation: Grid cell accuracy.** Reduction of grid cell accuracy was measured by calculating the fraction of spikes outside of the grid field boundaries divided by the total number of spikes. This value could be determined over the entire recording session or for particular time intervals with respect to light pulses. For excited grid cells (including both directly and synaptically excited cells), the accuracy of spikes between 0 and 10 ms after light onset was compared to the accuracy of an equal number of randomly selected spikes between 30 and 0 ms before light onset. For inhibited grid cells, a fixed time window could not be used because the pattern of inhibition was less stereotypic than for excitation. For each cell, a window of inhibition was thus selected that began with the minimum firing rate between 0 and 40 ms after light onset and ended when the cell regained a certain percentage of its baseline firing rate (e.g., 50%). The accuracy of all spikes during the inhibitory window (I) was compared to the accuracy of an equal number of randomly selected spikes between 30 and 0 ms before light onset (Pre).

**Measuring effects of optical stimulation: Distance to grid nodes.** The mean Euclidean distance of each spike from the closest grid field centroid was calculated for spikes during the I and Pre periods. For each cell, the Euclidean distance of an equal number of random $x$–$y$ coordinates from the path of the animal was also calculated to estimate chance distance estimates. To account for variability in grid field size, these values were normalized by the average grid spacing for each cell (i.e., average distance between the centroids of the cell's grid fields). By normalizing all Pre and I distances by the grid spacing, we could compare the relative change in the distance across grid cells, regardless of their field size. All values that included random sampling were calculated by taking the mean of 50 iterations.

**Measuring effects of optical stimulation: Speed tuning.** Speed tuning during stimulation sessions was calculated from a ±40 ms-wide PSTR. For inhibited speed cells, the window of maximal inhibition (I) was defined as starting from the time of the minimum in the firing rate until the cell regained to 50% of its baseline firing rate. For comparison, we also randomly extracted an equal number of spikes between 30 and 0 ms before light onset (downsampled-Pre). For each light pulse, we then performed a linear regression between the number of spikes occurring during I and the running speed closest to the video frame at the center of I as well as between the number of spikes occurring during downsampled-Pre and the

running speed closest to the video frame at 15 ms before every light trial. All values that included random sampling were calculated by taking the mean of 50 iterations.

**Measuring effects of optical stimulation: HD cell tuning**. Because narrow HD cells were generally not responsive to optical stimulation, HD tuning during optical stimulation sessions was calculated using fixed time windows. The HD score was generated for all spikes that occurred between 15 and 0 ms before light onset and for all spikes between 10 and 25 ms after light onset. The latter window was selected because it most closely matched the period when other cells showed maximal inhibition.

Because broad HD cells typically showed inhibition in response to optical stimulation, HD scores for broad HD cells were calculated for the period of maximal inhibition (from the minimum firing rate until the cell regained 50% of its baseline firing rate) and for an equal number of randomly selected between 30 and 0 ms before light onset. To ensure that downsampling spikes in the Pre condition did not lead to an artificial reduction in HD tuning, only cells that had a Pre HD tuning cutoff greater than their shuffled cutoff were included in the analysis (seven cells were excluded by this criterion). All values that included random sampling were calculated by taking the mean of 50 iterations.

**LFP analysis and theta modulation**. For each session, the LFP from the tetrode with the highest mean power between 6 and 10 Hz was selected for analysis. The raw LFP signal was downsampled to 1 kHz, and power at all frequencies was calculated using a Morlet wavelet. The average power between 6 and 10 Hz was defined as the theta amplitude.

To calculate theta modulation strength, the instantaneous theta phase was obtained from the Hilbert transform of the filtered signal between 6 and 10 Hz. Using spike and LFP timestamps, we linearly interpolated the firing phase for every spike. Theta modulation strength for each cell was defined as the mean resultant length of the distribution of firing phases. The same procedure was also used to assess spike phase preferences during stimulation. Similar to the analysis of grid and speed cells, the time window of maximum inhibition was identified for inhibited theta modulated cells (mean resultant length >0.1 in at least 2 out of the 3 sessions). Theta modulation strength and the phase angle were subsequently calculated for spikes during the inhibition window and for an equal number of randomly selected spikes between 30 and 0 ms before light onset. Modulation strength values that included random sampling were calculated by taking the mean of 50 iterations.

**Statistics**. All statistical tests were conducted using MATLAB r2009b or Graphpad Prism 6. The details of tests used are described with the results. For all comparisons of baseline versus stimulation, paired comparisons or repeated-measures tests were performed. Normality for all datasets was tested, and if the normality assumption was violated, an appropriate alternate test was used. For all tests involving a repeated-measures ANOVA, sphericity was not assumed and corrected for by using the Greenhouse–Geisser correction. All post hoc tests were performed by correcting for multiple comparisons. No statistical methods were used to pre-determine sample sizes, but our sample sizes are similar to those reported in relevant previous publications[24,30,61]. No blinding was used because blinding was not possible with our experimental procedures.

**Code availability**. The MATLAB code used in this study is available from the corresponding upon reasonable request.

## Data availability

The data that support the findings of this study are available from the corresponding author upon reasonable request.

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

## Acknowledgements

We thank G. de Guia for technical assistance with building microdrives. This work was supported by a Howard Hughes Medical Institute International Student Research Fellowship to I.Z., an Anandamahidol Foundation Fellowship to V.L., the Whitehall Foundation (20130571, 20140863), a Walter F. Heiligenberg Professorship to J.K.L., and by the National Institutes of Health (NS084324, NS086947, MH100349, MH108594, NS102915, NS097772 and MH107742).

## Author contributions

I.Z. and S.L. conceived and designed experiments. B.K.L., J.K.L. and S.L. provided expertise and feedback. V.L. and B.K.L. designed and provided viruses. I.Z. and M.L.F. collected data. I.Z. analyzed data. I.Z., J.K.L. and S.L. wrote the manuscript.

## Additional information

**Competing interests:** The authors declare no competing interests.

