## [Peer Review File · Nature Communications]

Reviewers' comments:

Reviewer #1 (Remarks to the Author):

Zutshi and collaborators have investigated in the present manuscript the contribution of layer II pyramidal (LIIP) neurons in the coding of various spatial and behavioral features as represented by head-direction (HD), grid and speed cells. To this end, they have selectively and optogenetically activated LIIP neurons and investigated the responses of the different classes of neurons. They have performed a very thorough analysis of their data, and report three main findings: grid cell and speed tuning are affected by this manipulation and HD cells (especially the ones showing narrow tuning) remain largely unaffected. However, the analyses of spatial tuning during optogenetic manipulation are statistically challenging, as pointed out by the authors themselves, raising concerns about the validity of the conclusions. The manuscript is very well written but it is not very clear how it contributes to our understanding of MEC circuits and information processing.

Major comments.

The most striking observation is that HD tuning is preserved when most other neurons are either excited or inhibited by LIIP neuron activation. However, a recent study manipulating the activity of interneurons in the MEC (Fig. 5 in Miao et al., 2017, cited by the authors) and another using chemogenetic manipulation (Kanter et al., 2017, cited by the authors) led to a very similar conclusion. It would have been interesting to show whether or not HD neurons in the MEC belong to a particular cell class, but, as other attempts from various labs before, it seems that there is no obvious relationship between behavioral correlates and genetically -defined cell classes.

Regarding the modulation of speed cells by the optogenetic manipulation, the statement that "Feedback inhibition from MEC superficial layer perturbation transiently disrupted speed tuning" (title of Fig. 3) is a bit misleading, as reported by the authors themselves. It is in fact difficult to statistically assess a change in speed score when basal firing rates are affected. A better method would have been to regress the firing rates on different spatial and speed features with a Generalized Linear Model (which would have taken into account the change in basal firing rates) but it is unlikely that this would give a positive result.

The same remark applies to the supposed change in grid cell accuracy. It is actually quite remarkable that despite a strong manipulation of the system, the grid tuning remains so accurate (unlike what is stated in the title of the manuscript). Yet, again, similar observations have already been reported.

Overall, it is surprising that the authors have limited their study to an optogenetic activation of LIIP neurons – which leads to a complex response of the network. An inhibition may have been much more informative regarding the role of this cell class in information processing in the MEC. It seems like the authors have used this particular stimulation protocol (short-lasting activation) to identify LIIP neurons and investigate their specific behavioral correlates. However, such optogenetic labeling is difficult to assess as the issues with isolating highly synchronous spikes usually prevent to really separate cells that are effectively driven directly by light stimulation from the ones that are entrained by network dynamics. Other types of protocol could have been used, like ramps or sinusoidal stimulation, which would have allowed a better discrimination of neurons directly entrained by light.

Minor comments:

L131: how many LIII interneurons were excited?

L162: Considering that speed cells seem to have very high firing rates, were they classified as interneurons? By the way, "Hz" is missing after average rate values.

Reviewer #2 (Remarks to the Author):

This manuscript addresses the effects of perturbations (via opto-genetic manipulations) of layer II (LII) processing in medial entorhinal cortex on the spatial tuning of different cell types in layers II-III in MEC. The authors perturb LII processing by directly and specifically stimulating LII pyramidal cells. They first demonstrate that LII pyramidal cells primarily project locally within LII MEC onto stellate cells, interneurons, and other LII pyramidal cells. They then activate these cells while monitoring speed, HD, theta, and grid cells. In general, they report that these manipulations disrupted the speed and grid cell signals while leaving HD and theta information intact. They conclude that speed tuning and grid information are processed locally within the superficial layers of MEC.

The manuscript was well-written and generally easy to follow. The experiments appear to be conducted thoroughly and the figures adequately demonstrate the findings. The statistical analyses are appropriate. The issue the paper addresses – namely how the speed and grid cell signals are constructed from known inputs and the local circuitry is an important one. The sophisticated genetic tools used here is also an excellent approach. While the results are convincing, I somewhat found them to be unsurprising – given what is already known about the local circuitry and inputs into MEC.

Previous anatomical studies have demonstrated that the pre/post-subiculum sends a massive projection to LIII MEC. This projection presumably conveys – at least in part – HD information to MEC. The authors frequently state that the HD signal is preserved in a separate MEC sub-network. However, given what we know about how the HD signal is processed, it is much more likely that the signal and its accompanying information is simply projected to MEC and it does not arise from a separate MEC sub-network as suggested by the authors. Therefore, perturbing LII MEC should not interfere with the HD signal – particularly because the HD signal is not solely being conveyed directly to LII pyramidal cells, but rather to cell types in other layers. Thus, the HD findings are not particularly surprising.

Similarly, given what is known about the septal projections to MEC, it could also be predicted that perturbing LII MEC would disrupt the speed signal. Justus et al. (2016) demonstrated that these septal projections convey speed-related information to MEC – specifically onto targets in LII – including pyramidal cells. Thus, based on this information, one would predict that perturbation of MEC LII pyramidal cell processing would interfere with the speed signal there.

Thus, this study, while using elegant state-of-the-art techniques is more confirmatory rather than breaking significant new ground.

Specific items:

Title: The title of the paper suggests that it has more to do with grid and HD cells, but the paper itself seems more focused on speed cells – or at least as much as on grid and HD cells.

Line 11: In the abstract, the last sentence would be better if it read: ‘..... while grid and speed information are dependent’

Line 44: The authors discuss the origin of the grid signal as a difference between local circuit computations vs. cellular processes. While it is clear what is meant by local circuit computations, it is not clear what is meant by ‘cellular’, and how this is different from the local circuit hypothesis.

Line 68: ‘.....and then optogenetically activated LIIP cell to recruit LII cell excitation.....’. By LII cell excitation, I presume the authors mean LII stellate cells and interneurons. If so, this should be

stated.

Line 86: 'This virus allowed us to label cell bodies and axons with the' Cell bodies and axons of what? All cells? LIIP cells? Please specify.

Line 92: Define a.u.

Line 158: The rationale for using two different stimulation frequencies – 8 and 12 Hz – should be explained.

Pg 7-9: It is not clear how long the optical stimulation lasted (eg., line 110). Was it the entire session or just intermittent? This should be discussed in the context that while overall firing rates for the entire session appeared unaffected, firing rates were affected on the short time scale of 25 msec.

Line 356: '50% are stellate cells' needs a citation.

Page 18, top paragraph: The authors bring up the possibility that narrow HD cells may be controlled by a subset of functionally distinct interneurons. However, as stated above, given what we know about the HD circuit, it seems more likely that the MEC just inherits the signal/information from upstream structures.

Reviewer #3 (Remarks to the Author):

The paper checked whether the computations sustaining speed, grid and head-direction cells are dependent on local computations of MEC superficial layers. They used optogenetics to manipulate pyramidal cell in layer 2, which project to the local circuitry. Their readout was the activity of cells in layer 2\3 in MEC.

They found that activation of layer 2 pyramidal cells led to strong perturbation of cellular activity within layers 2\3. Some neurons acted with excitation, some with inhibition, and some with a combination of both. The optical stimulation affected grid and speed cells. However, narrow head-direction cells were much less affected. This suggests that the head-direction signal is inherited from other areas, in contrast to the grid and speed signals, which may be computed locally.

The paper has many nice results. The main problem with the paper is that it describes many phenomena, without thinking of some system to connect those phenomena together. The paper, while showing some compelling data, would have benefitted from some more thought trying to understand how these different cell types and phenomenological responses would fit into some coherent system. It would have also helped the reader to follow all the details, if such an attempt to generate some common narrative would have been made.

Major comments:

1. Fig. 2c – The cells are divided into different categories. How were the categories determined? This is not completely clear from the paper. Were they forming a continuum, or rather arbitrary divisions? How much are the results dependent on the thresholds used for categorization?
2. Fig 2c - Some of the "No response" examples seem to contain a small inhibitory response that cannot be neglected, and is reflected in the summed population activity (red trace). This same issue is seen in Fig. 5c – while it is stated in the text that there is no response, clearly there is a small inhibitory response in many of the cells, which is reflected also in the population activity.
3. Fig. 2f – the clustering of interneurons should be done while taking into account both axes, and not only firing rate. For example, 4 of the gray diamonds (and some other cells there) seem to belong to

the excitatory neuron cluster on the left, while some of the pink dots with a low firing rate (those with a peak-valley ratio below about 0.7) actually clearly belong to the interneuron cluster. Please re-cluster the cells and see how it affects the results. It seems that it will only strengthen the paper's findings.

4. Fig. 4c – The authors seem to find a potential resonance effect at about 8 Hz. However they have not tried a lower frequency of stimulation (say at 4 or 6 Hz), so they should reserve their claim about 8 Hz, as it could have been that the stronger effect would have been found at an even lower frequency of stimulation.

5. Suppl. Fig 4b – While the narrow HD-cells form a cluster, the wide-HD cells seem to form a continuum with unclassified cells (The blue and black dots are not separable as clusters). How does changing the criterion to a 99-percentile (for this a shuffling of 1000 will be required) change your results?

6. The Discussion part does not attempt to try to connect the different phenomena together, and provide some explanation as to why these phenomena occur the way they do in the different subsets of cells. Furthermore, the discussion trying to connect the phenomena to the thalamus is very weak and farfetched, and should be removed.

Minor comments:

1. It is better to change the title to include also the speed component: "OPTOGENETICALLY INDUCED FEEDBACK INHIBITION OF MEDIAL ENTORHINAL CORTEX SUPERFICIAL LAYERS TRANSIENTLY PERTURBS GRID ACCURACY BUT NOT HEAD DIRECTION and speed TUNING"

2. Fig. 1e – The distribution does not seem Gaussian, so it is better to use a non-parametric t-test (Same comment for Suppl. Fig. 1c,d).

3. Fig. 1g – the authors should elaborate a bit more, for the sake of clarity, how the image explains their claim?

4. Fig. 2c – For readability, please sort the cells in the raster plots according to the estimated latencies of the response.

5. Fig. 2f – ratios should be plotted on a logarithmic axis. For example, the peak-valley ratio in the y-axis of this panel should be logarithmic. Also the scale (and calculation of errors) in the y-axis of fig. 2g should be logarithmic for the same reason.

6. Fig. 2f – the color legend in panel e does not completely match the colors of the points. There is no gray in the legend, so it is not completely clear if the gray means only inhibition or no response.

7. Fig. 3 – It is not clear if any of the speed cells shown in this figure were classified as interneurons

8. Fig. 4d and 4e – The graphs for 8 Hz should be shown too.

9. Supp Fig.1a – "the damage observed in the sections is caused by postmortem tetrode extraction". Could the implantation itself not be a reason too?

10. Suppl. Fig. 2b – The measure of Peak is very noisy, and highly depends on the level of smoothing. How does this classification depend on the smoothing of the signal?

11. Suppl. Fig. 2j is important. It should be moved to a main figure.

12. Suppl. Fig. 4b – please make the figures square (that is "axis equal"). Also, please show a histogram of the HD score, to emphasize the bi-modality. In addition, it is not clear what are black dots doing above the diagonal? I thought this figure was used for the determination of the colors of the dots.

13. Supplementary Figure 5 – " The phase angle preference was preserved across sessions." While generally true, the "Baseline 2" map does not show a very good recovery of tuning of the cells.

RESPONSE TO REVIEWERS, NCOMMS-17-28893-T

We thank the three reviewers for their thorough evaluation of our manuscript and insightful comments. All reviewer comments are comprehensively addressed as described in our point-by-point responses, where we retained reviewer comments in *black font* and added our replies in *blue font*. In following the advice of the editor, we preceded the point-by-point responses with a section that more broadly speaks to concerns of conceptual advance.

Response to concerns of conceptual advance

Reviewers #1 and #2 expressed concerns about the conceptual advance of our work compared to previous work. We identify at least two key points for which we consider our findings a substantial advance beyond previously published work.

(1) Evidence that computations within the mEC superficial layers determine the grid location

Although not all previous similar work is explicitly listed by the referees, we gather that the most relevant publications for comparison to our work are those that targeted intra-entorhinal circuits: (1) the optogenetic activation of entorhinal PV cells by Buetfering et al., 2014, (2) the chemogenetic inhibition of entorhinal PV and SOM cells by Miao et al., 2017, (3) the chemogenetic inhibition and excitation in ~27 % of layer II stellate cells by Kanter et al., 2017, (4) the NMDA receptor deletion in retrohippocampal regions by Gil et al., 2018, and (5) the HCN1 channel knockout in mEC by Mallory et al., 2018. We apologize if any relevant publication is omitted from this list and would ask the referees to identify the publication so that we are in a position to directly respond to specific concerns.

-Of the listed publications, Kanter et al., 2017 is the only one that selectively targeted the superficial layers of mEC and they explicitly concluded that their manipulation only altered firing rates but not the firing location of grid fields. The other four publications differ from ours in that they did not selectively target specific entorhinal layers. Although each study found a different pattern of effects on grid cells, none of their results resembled ours of a selective and transient effect on grid accuracy. First, Buetfering et al., 2014 did not report any effect on grid patterns with PV cell activation. Second, Miao et al., 2017 did not find effects on grid cells with SOM inactivation and predominantly observed additional spiking between grid nodes rather than a shift in the location of the grid peaks with PV inactivation. In their discussion, they explicitly state 'It is worth noting that the inactivation of the PV interneurons did not completely abolish the structure of the grid pattern. While the hexagonal firing pattern was blurred by enhanced firing between the grid fields, the average locations of the firing vertices were mostly retained, resulting in low, but non-zero, grid scores. Such residual hexagonality might be expected, for example, if the tuning of the speed cells is only partly abolished, as in the present data. Grid patterns might also be maintained by recurrent pathways in other subcircuits of the mEC, involving connections via other subtypes of interneurons, or direct excitatory connections between pyramidal cells, in layer II or III (Dunn et al., 2015; Tocker et al., 2015).' **It is our work that is the first to experimentally address the latter possibility and to**

demonstrate a role of the superficial layers in determining the location accuracy of grid patterns.

More comprehensive effects on grid patterns with local manipulations have recently been described by Gil et al., 2018 and Mallory et al., 2018, who deleted NMDA receptors from all retrohippocampal areas or HCN channels from all mEC cells. While their studies clearly show a role of mEC and/or the surrounding cortices in generating grid patterns, their manipulations were either targeted to dendritic processing or cellular resonance of all mEC cells, which precludes that they can distinguish between the contribution of local connections within mEC and effects of processing afferent inputs, which have in many previous studies been shown to result in a disruption of grid patterns (Bonnievie et al., 2013; Brandon et al., 2011; Koenig et al., 2011; Winter et al., 2015). Furthermore, it is also relevant to note that Gil et al. reported an expansion rather than disruption of grid patterns

Taken together, our study is therefore distinct from each of these previous publications, in that it implicates recurrent connectivity within the entorhinal cortex in grid firing patterns. This finding is important because it is the first to experimentally confirm an explicit or implicit assumption of most major theories of grid cell generation.

While our effect size on inaccurate grid spike locations (1.14, see Fig. 7k) may be judged to be relatively small (~22 % on the scale from 1 [unchanged grid pattern] to 1.65 [random locations on the path]), an additional novel finding of our study is also that the spatial accuracy of grid firing is rapidly corrected. Without the rapid correction, a misalignment on the observed scale would – only after a few cycles of our manipulation – result in a completely relinquished grid pattern.

(2) Evidence that narrow and broad HD-selective cells form functionally segregated subcircuits within mEC

With regard to HD selectivity, we agree that our findings are replicating the widely reported result that head direction coding persists while grid patterns are disrupted (e.g., Bonnievie et al., 2013; Brandon et al., 2011; Koenig et al., 2011; Miao et al., 2017). While our results are of course confirmatory of all of these previous findings, they reveal insight beyond a mere dissociation of grid and HD coding. We find that there is also a dividing line between broad/conjunctive HD cells and the remaining narrowly tuned HD cells within the mEC circuit. The former respond in the same way as grid and speed cells, and the latter form a functionally segregated population. Importantly, we find that the two intermingled cell populations (i.e., narrow HD and all other mEC cell types) can form completely separate subcircuits within the same cortical region.

Response to concerns of reviewer #1.

Reviewer #1 (Remarks to the Author):

Zutshi and collaborators have investigated in the present manuscript the contribution of layer II pyramidal (LIIP) neurons in the coding of various spatial and behavioral features as represented by head-direction (HD), grid and speed cells. To this end, they have selectively and optogenetically activated LIIP neurons and investigated the responses of the different classes of neurons. They have performed a very thorough analysis of their data, and report three main findings: grid cell and speed tuning are affected by this manipulation and HD cells (especially the ones showing narrow tuning) remain largely unaffected. However, the analyses of spatial tuning during optogenetic manipulation are statistically challenging, as pointed out by the authors themselves, raising concerns about the validity of the conclusions. The manuscript is very well written but it is not very clear how it contributes to our understanding of MEC circuits and information processing.

While the analysis of data from manipulations of complex neural circuits presents challenges, the reviewer appears to ascertain that we were as thorough as conceivable in addressing these challenges. While specific comments of reviewers on the data analysis are addressed in the point-by-point responses that follow, we would like to iterate that all of our analysis included key statistical controls to assure the validity of our data analysis and conclusions. Examples of the attention to detail include: (1) Analysis for the inaccuracy of grid firing was performed by comparison of spike locations to random locations along the animals' path (rather than any random location). (2) We used two separate analysis methods to confirm that grid location was indeed transiently disrupted. (3) Perhaps most importantly, we meticulously controlled for any effects of firing rate by performing all comparisons against samples from the prestimulation period that were precisely matched for spike numbers. Here, we of course observed that some calculations were dependent on firing rates (e.g., HD selectivity increases with low spike numbers, speed tuning decreases with low spike numbers) but comparisons against matched samples nonetheless yielded the result that grid firing locations were disrupted while HD and speed tuning were altered only to the extent predicted from the reduced spike numbers.

In addition, we were also conscientious to be fully up front in the manuscript about the limitations of our analysis with regard to the neural code for running speed. Because the speed code of mEC cells is a rate code, it is possible that reduced firing rates preclude that the code is effectively read out by grid cells. However, if the effect on grid cells were directly related to the disruption in speed coding, we would expect to see that the firing of all grid cells is disrupted to an approximately equal extent. However, we found that grid accuracy showed the most pronounced disruption in grid cells with the strongest feedback inhibition (see Fig. 7I), which is an indication that processing at the level of grid cells or the grid cell network contributed to the effect.

These points are now discussed in substantially revised paragraphs in the discussion (line 369-414). Furthermore, how our findings contribute to the understanding of the mEC circuit is also mentioned in more detail in the rewritten abstract (line 1-12) and in the last paragraph of the discussion (line 439-450: *'Taken together, we find that narrow HD signals form a separate subcircuit within the superficial layers, which points to a network mechanism for separate local processing of afferent HD signals that could explain the persistence of HD signals during*

manipulations that alter grid firing^{20,21,26}. Furthermore, our results suggest that computations within recurrent circuits in the superficial layers that consists of broadly tuned HD cells, speed cells, grid cells and/or cells that are conjunctive for these properties are critical for determining the precise firing location of grid cells. While processing of afferent information by these functional cell types and recurrent circuits between grid cells have been the standard tenet of computational models for grid firing^{13,14}, our results are a critical experimental confirmation that key computations for grid accuracy are performed within the superficial layers. Finally, our results provide evidence that, under circumstances of foraging with abundant sensory cues, any local perturbations of position accuracy are corrected on a timescale in the order of milliseconds.').

Major comments.

The most striking observation is that HD tuning is preserved when most other neurons are either excited or inhibited by LIIP neuron activation. However, a recent study manipulating the activity of interneurons in the MEC (Fig. 5 in Miao et al., 2017, cited by the authors) and another using chemogenetic manipulation (Kanter et al., 2017, cited by the authors) led to a very similar conclusion. It would have been interesting to show whether or not HD neurons in the MEC belong to a particular cell class, but, as other attempts from various labs before, it seems that there is no obvious relationship between behavioral correlates and genetically-defined cell classes.

We agree that there has been a long-standing record of finding that head direction coding is preserved during manipulations of mEC that alter grid firing, which is the case even for manipulations that find more substantial effects on grid firing than those in Miao et al., 2017 and Kanter et al., 2017 (e.g., Bonnevie et al., 2013; Brandon et al., 2011; Koenig et al., 2011). However, unique about our data is the finding that we do not only find a preservation of HD coding but that we also find that narrowly-tuned HD cells form a local network that is separate from the broader HD cells and cells with conjunctive HD tuning (e.g., grid-by-HD cells). As also stated above, our findings therefore go beyond showing a mere dissociation between grid and HD coding. We also identify a functional segregation between HD cells with different degrees of HD coding, and the segregation may contribute to HD tuning being resilient to manipulations that affect grid cells. Our data thus add to the well-known observation that sharp HD tuning is received from afferent input and establish that the sharply tuned afferent HD input is sustained within mEC by a separate sub-circuit.

Regarding the modulation of speed cells by the optogenetic manipulation, the statement that “Feedback inhibition from MEC superficial layer perturbation transiently disrupted speed tuning” (title of Fig. 3) is a bit misleading, as reported by the authors themselves. It is in fact difficult to statistically assess a change in speed score when basal firing rates are affected. A better method would have been to regress the firing rates on different spatial and speed features with a Generalized Linear Model (which would have taken into account the change in basal firing rates) but it is unlikely that this would give a positive result.

We apologize that the title of the figure was misleading and that it was not as explicit as intended in reporting that speed coding was only reduced to the extent that was predicted by the reduced spike numbers during inhibition. We therefore changed the title of Fig. 3 (Now: *'Speed cells were transiently inhibited by LIIP stimulation.'*) to better convey that the primary effect is on firing rates and that speed tuning is disrupted as a consequence of the effects of our manipulation on basal firing rates. With regard to the comment on the GLM analysis we agree with the view that this would likely not have changed our results. However, if the reviewer would like us to use this method, we would appreciate further guidance on how to implement the analysis and, in particular, to more specifically identify the spatial and speed features that are referred to in the comment.

Furthermore, to make sure that our results on speed coding are not misunderstood, we rewrote the concluding statements of the section on speed tuning (Line 192-198: *'Speed scores did not differ between the I window and the downsampled Pre window, but were in both cases reduced compared to baseline, suggesting that subsampling to low firing rates is sufficient to lower the score (Fig. 3e). Indeed, the speed scores during I were correlated to the number of spikes in the window (Fig. 3f), and speed scores rapidly increased as cells emerged from inhibition (Fig. 3g). Therefore, the decrease in speed tuning of mEC speed cells did not exceed the level that could be explained by reduced spiking during the inhibition window.'*).

The same remark applies to the supposed change in grid cell accuracy. It is actually quite remarkable that despite a strong manipulation of the system, the grid tuning remains so accurate (unlike what is stated in the title of the manuscript). Yet, again, similar observations have already been reported.

We share the view of the reviewer that grid tuning is remarkably resilient to strong manipulations of local entorhinal circuits, which has also been the findings of previous studies by Buetfering et al., 2014, Miao et al., 2017 and Kanter et al., 2017. Given the resilience of the firing patterns to local manipulation, our finding of even a moderate and transient inaccuracy in grid firing therefore indicates a contribution of local circuits in the superficial layers. Because optogenetic manipulations allow for analysis on a faster time scale than chemogenetic manipulations, we were also able to establish that error correction is rapid, which may explain why all previous studies as well as our own have not observed a cumulative effect on grid cells that would result in a large-scale disruption of grid patterns.

In addition, we also consider our experimental evidence of remarkably fast computations in the grid circuit as potentially paradigm-shifting for conceptual and computational models of grid cells. While the emphasis has in the past been on a network that can internally sustain the location information over long time intervals, our findings indicate that the available afferent information makes a substantial contribution on a short time scale. For example, the local network could be strongly prewired to rapidly generate grid patterns as long as the appropriate afferent information is available. This interpretation would be consistent with published studies that have so far revealed strong effect on grid patterns with manipulations of afferent projections (Bonnievie et al., 2013; Brandon et al., 2011; Koenig et al., 2011; Winter et al., 2015).

Overall, it is surprising that the authors have limited their study to an optogenetic activation of LIIP neurons – which leads to a complex response of the network. An inhibition may have been much more informative regarding the role of this cell class in information processing in the MEC. It seems like the authors have used this particular stimulation protocol (short-lasting activation) to identify LIIP neurons and investigate their specific behavioral correlates. However, such optogenetic labeling is difficult to assess as the issues with isolating highly synchronous spikes usually prevent to really separate cells that are effectively driven directly by light stimulation from the ones that are entrained by network dynamics. Other types of protocol could have been used, like ramps or sinusoidal stimulation, which would have allowed a better discrimination of neurons directly entrained by light.

As presented in our manuscript, we began our work with a detailed anatomical description of LIIP cell projections, which led to the finding of prominent local axon terminals within ipsilateral LII. Although connections of LIIP cells to other cells in the superficial layers have been described in slice studies (e.g., Couey et al., 2013; Fuchs et al., 2016; Winterer et al., 2017), our anatomical tracing first reports that entorhinal LIIP synapses are exclusively within layer II. Based on these anatomical findings and our preliminary electrophysiological data in freely behaving mice, we were well aware that the highly locally recurrent nature of LIIP cells would be prohibitive for any attempts to use standard optogenetic tagging methods to distinguish between directly driven and network-driven responses of LII cells. Our reasoning for nonetheless describing data analysis that could potentially discriminate between direct and indirect responses (line 992-1014) was to confirm our assumption that these responses could not be separated.

However, we realized that the dense local connectivity of LIIP cells may afford us the opportunity to study circuits in the superficial layers in a similar way as has previously been done with electrical stimulation of the recurrent CA3 networks. Here, brief electrical stimulation of the axons of CA3 cells in the hippocampal commissure resulted in excitation followed by inhibition, and this method has been used to examine whether ongoing hippocampal activity is necessary for phase precession and for place coding (Moser et al., 2005; Zugaro et al., 2005) and to briefly inhibit ongoing activity during sharp wave ripples (e.g., Ego-Stengel and Wilson, 2010; Jadhav et al., 2012).

We therefore tested whether the dense recurrent connectivity of LIIP cells would similarly result in a period of feedback inhibition in response to stimulation. We observed the predicted pattern in mEC superficial layers and used it as an effective strategy to transiently disrupt neuronal activity in the superficial layers. In addition, we were initially considering to pursue the suggested strategy of direct transient inhibition of LIIP cells. However, as our results began to reveal that even inhibiting a substantial fraction of all LII and LIII cells (i.e., during the feedback inhibition after stimulation) did not have any effect on HD coding and had only transient, moderate effects on grid coding, we reasoned that further scaling back our inhibition to a smaller population of cells in the superficial layers would not yield positive results.

Minor comments:

L131: how many LIII interneurons were excited?

After reclassifying interneurons based on the recommendation by reviewer 3, we only identified 4 putative interneurons from LIII. Of these 4, one LIII interneuron was excited. However, we note that the cell bodies of interneurons that mediate the feedback inhibition in LIII do not need to be located within LIII. Given that the cell location is not particularly relevant and that the low number of interneurons in LIII does not allow for an accurate estimate of the fraction of responding neurons, we therefore now report the total number of LII/III interneurons that respond with excitation (23 of 30, line 138). We continue to identify LII and LIII interneurons with different symbols in Fig. 2f (circles for layer II and diamonds for layer III, color-coded for inhibition/excitation according to the legend in 2e).

L162: Considering that speed cells seem to have very high firing rates, were they classified as interneurons? By the way, “Hz” is missing after average rate values.

Thank you for pointing out that the unit was missing. ‘Hz’ is now added (line 173).

As in the publication that originally identified speed cells in mEC (Kropff et al., 2015) and in a recent publication that identified most speed cells as fast spiking (Ye et al., 2018), we included interneurons. We now explicitly report the fraction of speed cells that are interneurons (i.e., 15 of 47, 31.9 %) in the main text (line 173) and figure legend (Figure 3, line 675). In addition, we provide an updated detailed breakdown of cell type, speed score, and firing rate in Supplementary Figs. 4f and g.

Response to concerns of reviewer #2.

Reviewer #2 (Remarks to the Author):

This manuscript addresses the effects of perturbations (via opto-genetic manipulations) of layer II (LII) processing in medial entorhinal cortex on the spatial tuning of different cell types in layers II-III in MEC. The authors perturb LII processing by directly and specifically stimulating LII pyramidal cells. They first demonstrate that LII pyramidal cells primarily project locally within LII MEC onto stellate cells, interneurons, and other LII pyramidal cells. They then activate these cells while monitoring speed, HD, theta, and grid cells. In general, they report that these manipulations disrupted the speed and grid cell signals while leaving HD and theta information intact. They conclude that speed tuning and grid information are processed locally within the superficial layers of MEC.

The manuscript was well-written and generally easy to follow. The experiments appear to be conducted thoroughly and the figures adequately demonstrate the findings. The statistical analyses are appropriate. The issue the paper addresses – namely how the speed and grid cell signals are constructed from known inputs and the local circuitry is an important one. The

sophisticated genetic tools used here is also an excellent approach. While the results are convincing, I somewhat found them to be unsurprising - given what is already known about the local circuitry and inputs into MEC.

We agree that some aspects of our results are unsurprising because they were previously shown (e.g., preserved HD coding). However, other results may also appear unsurprising because they correspond to widely held assumptions, but nonetheless assumptions that had not been substantiated by previous experimental data. One such finding, as also pointed out by the reviewer, is that a selective manipulation of only the superficial mEC layer results in spatially inaccurate firing patterns of grid cells. This result is to our knowledge the first experimental evidence that grid accuracy is reduced by selectively disrupting neural computations in the superficial layers. More detailed comparisons with previously published findings are included in the general response to the concerns by reviewers #1 and #2.

Previous anatomical studies have demonstrated that the pre/post-subiculum sends a massive projection to LIII MEC. This projection presumably conveys – at least in part – HD information to MEC. The authors frequently state that the HD signal is preserved in a separate MEC sub-network. However, given what we know about how the HD signal is processed, it is much more likely that the signal and its accompanying information is simply projected to MEC and it does not arise from a separate MEC sub-network as suggested by the authors. Therefore, perturbing LII MEC should not interfere with the HD signal – particularly because the HD signal is not solely being conveyed directly to LII pyramidal cells, but rather to cell types in other layers. Thus, the HD findings are not particularly surprising.

We apologize if any of our writing gave the impression that we suggest that the HD signal arises from within mEC, which we believe is distinct from stating that there is a separate local network of sharply tuned HD cells within mEC. We are well aware of the major projection from pre/postsubiculum to mEC and completely agree that the HD signal in mEC arises from these projections. Yet, the new finding of our study is that some of the target neurons of the HD signal within mEC are not under the same inhibitory control as the local network that includes speed cells, broadly tuned HD cells, grid cells, and conjunctive cells.

One possible interpretation of our results is therefore that the HD network that is afferent to layer III sustains a local HD network in mEC cell layers, which then contributes – in addition to direct afferent HD information – to making the HD signal broadly available within mEC, for example to the broad HD cells that appear integrated with the grid/speed coding cells. The local sharp HD network in mEC thus appears to function as an additional pool of narrow HD cells that are anatomically but not functionally intermingled with other cell types.

To address these points in the manuscript and make sure that we are explicit in stating that HD information is afferent, we completely rewrote the relevant part of the discussion (line 349-367) and now lead in with the phrase, *'It is well established that HD tuning is forwarded via the anterior thalamic nucleus and postsubiculum to mEC^{15,16,48}, ...'*.

Similarly, given what is known about the septal projections to MEC, it could also be predicted that perturbing LII MEC would disrupt the speed signal. Justus et al. (2016) demonstrated that these septal projections convey speed-related information to MEC – specifically onto targets in LII – including pyramidal cells. Thus, based on this information, one would predict that perturbation of MEC LII pyramidal cell processing would interfere with the speed signal there.

We agree that a disruption of speed coding by manipulating the LIIP cells was expected based on the elegant findings by Justus et al., 2016 and the finding that speed coding is more predominant in LIIP cells (Sun et al., 2015). Furthermore, the speed code is a rate code and would thus be expected to be perturbed by any manipulation that has direct effects on firing rates. However, we nonetheless did not find that speed coding was entirely disrupted but only reduced to the extent that could be predicted from reduced spike numbers during the period of inhibition. These findings are consistent with the interpretation that the speed signal is afferent to mEC (as predicted from Justus et al) but not substantially further processed within mEC (our finding). However, unlike for HD coding, we did not find a separate local network of speed cells that was separate from the local grid network. To better convey the complexity of the results on speed cells, we changed the subtitle and rewrote the section on speed cells in the discussion (*'Integration of speed tuning in mEC networks'*, line 369-393). In addition, we also updated the concluding statements of the Results section on speed tuning (Line 192-198: *'Speed scores did not differ between the I window and the downsampled Pre window, but were in both cases reduced compared to baseline, suggesting that subsampling to low firing rates is sufficient to lower the score (Fig. 3e). Indeed, the speed scores during I were correlated to the number of spikes in the window (Fig. 3f), and speed scores rapidly increased as cells emerged from inhibition (Fig. 3g). Therefore, the decrease in speed tuning of mEC speed cells did not exceed the level that could be explained by reduced spiking during the inhibition window.'*). Given that these results are consistent with the prediction that speed signals are afferent but also show that local speed coding is preserved at the level that can be predicted when accounting for rate reduction, we would ask for consideration that the results were not entirely predictable.

Thus, this study, while using elegant state-of-the-art techniques is more confirmatory rather than breaking significant new ground.

While we agree that our study does not contradict any of the previous findings, we would nonetheless advocate that our study does not only use state-of-the-art techniques and analysis, but also breaks new ground in experimentally demonstrating the widely held assumptions that local circuits within the superficial layers are critical for grid accuracy. In addition, our finding that local disruptions can be rapidly corrected also provides new constraints on how recurrent local networks and afferent inputs contribute to computations within the grid cell network. Furthermore, we do not only show that HD information is preserved, but also that HD information is partially separate while speed information is fully integrated with the grid network. These points are also addressed in the response to concerns about conceptual advance.

Specific items:

Title: The title of the paper suggests that it has more to do with grid and HD cells, but the paper itself seems more focused on speed cells – or at least as much as on grid and HD cells.

In response to the comment and as suggested by reviewer #3, we changed the title to *'OPTOGENETICALLY INDUCED FEEDBACK INHIBITION OF MEDIAL ENTORHINAL CORTEX SUPERFICIAL LAYERS TRANSIENTLY PERTURBS GRID ACCURACY BUT NOT HEAD DIRECTION AND SPEED TUNING'*. In response to the following comment, we also edited the abstract (line 1-12) to address the concern that speed cells are not sufficiently mentioned.

Line 11: In the abstract, the last sentence would be better if it read: '..... while grid and speed information are dependent

Because speed tuning is rate-coded and we analyzed the period of inhibition, we cannot unequivocally argue that there is an effect on speed tuning. Rather, we observed that the tuning curves of speed cells were reduced, but only to the extent that could be predicted from the reduced spiking. We apologize that our original version was not sufficiently clear in disambiguating these results on speed cells. We have now rephrased these sections of the abstract, which now read (line 6-12):

'In contrast, speed, broadly tuned HD cells, and grid cells showed pronounced transient excitatory and inhibitory responses, with a reduction in grid accuracy during the period of feedback inhibition. The recovery of these cells to baseline firing rates within approximately 25 milliseconds was accompanied by a concurrent correction in grid tuning. These results suggest that sharp HD cells are embedded in a separate mEC sub-network from broad HD cells, speed cells, and grid cells and that grid tuning is dependent on local processing, but also rapidly updated by HD, speed, or other afferent inputs to mEC circuits.'

To also address these points in more detail, but without the space limitations of the abstract, we also rewrote the corresponding section in the discussion (*'Integration of speed tuning in mEC networks'*, line 369-393).

Line 44: The authors discuss the origin of the grid signal as a difference between local circuit computations vs. cellular processes. While it is clear what is meant by local circuit computations, it is not clear what is meant by 'cellular', and how this is different from the local circuit hypothesis.

We meant that dendritic processing and cellular resonance may differ between anatomical cell types (e.g., stellate cells or pyramidal cells). This conjecture may only have been accessible from statements later in the paragraph and we therefore now replaced *'cellular rather than local circuit computations'* with *'dendritic processing or resonance properties of a cell'*.

The entire sentence (line 43-47) now reads:

'The limited effects of local circuit manipulations on grid cells therefore raise the possibility that dendritic processing or resonance properties of a cell predominantly contribute to grid generation, and that grid firing may thus selectively emerge in a particular morphological cell type. Numerous studies have therefore compared the two major morphological cell types in mEC layer II – stellate (LIIS) and pyramidal cells (LIIP).'

Line 68: '.....and then optogenetically activated LIIP cell to recruit LII cell excitation.....'. By LII cell excitation, I presume the authors mean LII stellate cells and interneurons. If so, this should be stated.

Yes, we meant that excitation could be elicited directly in LIIP cells and indirectly in LIIS and that the synchronous activation of both excitatory cell classes would result in the recruitment of interneurons, which mediate the feedback inhibition. This is now stated (line 66-70, *'We therefore first confirmed that LIIP cells have the majority of their synaptic terminals within mEC superficial layers³⁶ and then optogenetically activated LIIP cells to recruit LIIS cell excitation, and as a consequence of direct and indirect activation of both principal cell types, interneuron driven feedback inhibition that transiently perturbed network activity across layers II and III.'*).

Line 86: 'This virus allowed us to label cell bodies and axons with the'. Cell bodies and axons of what? All cells? LIIP cells? Please specify.

LIIP cells. Now specified (now line 87).

Line 92: Define a.u.

Now converted to mm² (now line 93).

Line 158: The rationale for using two different stimulation frequencies – 8 and 12 Hz – should be explained.

Now explained (line 114-118: *'The two distinct frequencies were chosen to control for the possibility that there might be a frequency dependent effect of rhythmic excitation of LIIP cells, especially for frequencies close to theta oscillations (~8 Hz). Moreover, both of these frequencies allowed sufficient time for recovery of cells between light pulses (83.3 ms for 12 Hz, 125 ms for 8 Hz), but also regularly perturbed the network throughout the session.'*).

Pg 7-9: It is not clear how long the optical stimulation lasted (eg., line 110). Was it the entire session or just intermittent? This should be discussed in the context that while overall firing rates for the entire session appeared unaffected, firing rates were affected on the short time scale of 25 msec.

We now include the information on the duration of the optical stimulation and describe the experimental design in more detail (line 113-114: *'Throughout the entire stimulation session, LIIP cells were repetitively activated at one of two stimulation frequencies – 8 Hz or 12 Hz.'* and

line 167-170: ‘We began our analysis with speed cells and compared 10-15 minute sessions with repetitive light stimulation (delivered at 8 Hz or 12 Hz for the entire session duration) to 10-15 minute baseline sessions (without light stimulation) that occurred before and after the light stimulation sessions.’).

While we did not report average firing rates over the entire session in the original manuscript, we now calculated these values for the different groups of cells (Only excitation, Excitation and inhibition, Only inhibition, and No/weak response) and show them in the plot below. Excitation only resulted in a non-significant trend for an increase in the firing rate of cells, presumably due to the extremely short duration of the excitation. However, inhibition led to a minor decrease in firing rate for the ‘Excitation and inhibition’ group and for the ‘Only inhibition’ group, consistent with there being a ~25 ms long effect on firing rate following light stimulation (Excitation and inhibition: One-way repeated measures ANOVA followed by Tukey’s multiple comparisons test, $n = 61$ cells from 7 mice, $F(1.966, 117.9) = 5.21, p = 0.0071$, Only inhibition: One-way repeated measures ANOVA followed by Tukey’s multiple comparisons test, $n = 104$ cells from 7 mice, $F(1.92, 197.8) = 4.925, p = 0.0090$). Because these effects are exactly as expected, we considered that the panel below would not add substantially to the information that is presented and did not include the panel in the manuscript. We would of course follow the advice of the reviewer if it is recommended to add the panel.

Line 356: ‘50% are stellate cells’ needs a citation.

Citation is added (now line 359). The citation is:

Gatome, C. W., Slomianka, L., Lipp, H. P. & Amrein, I. Number estimates of neuronal phenotypes in layer II of the medial entorhinal cortex of rat and mouse. *Neuroscience* 170, 156–165 (2010).

Page 18, top paragraph: The authors bring up the possibility that narrow HD cells may be controlled by a subset of functionally distinct interneurons. However, as stated above, given what we know about the HD circuit, it seems more likely that the MEC just inherits the signal/information from upstream structures.

We did not consider these possibilities mutually exclusive and now specifically state that

narrow HD cells inherit the information and may then be locally controlled by a distinct set of interneurons (line 350-353).

'It is well established that HD tuning is forwarded via the anterior thalamic nucleus and postsubiculum to mEC^{15,16,48}, and we find that afferent HD information is retained in an entorhinal sub-network of sharply tuned HD cells with limited functional connectivity to the majority of principal cells and interneurons.'

In addition, we also emphasized this point in the concluding paragraph (line 439-441: *'Taken together, we find that narrow HD signals form a separate subcircuit within the superficial layers, which points to a network mechanism for separate local processing of afferent HD signals that could explain the persistence of HD signals during manipulations that alter grid firing^{20,21,26}.'* and represented the information in a schematic that is now added (Supplementary Fig. 8).

Response to concerns of reviewer #3.

Reviewer #3 (Remarks to the Author):

The paper checked whether the computations sustaining speed, grid and head-direction cells are dependent on local computations of MEC superficial layers. They used optogenetics to manipulate pyramidal cell in layer 2, which project to the local circuitry. Their readout was the activity of cells in layer 2\3 in MEC.

They found that activation of layer 2 pyramidal cells led to strong perturbation of cellular activity within layers 2\3. Some neurons acted with excitation, some with inhibition, and some with a combination of both. The optical stimulation affected grid and speed cells. However, narrow head-direction cells were much less affected. This suggests that the head-direction signal is inherited from other areas, in contrast to the grid and speed signals, which may be computed locally.

The paper has many nice results. The main problem with the paper is that it describes many phenomena, without thinking of some system to connect those phenomena together. The paper, while showing some compelling data, would have benefitted from some more thought trying to understand how these different cell types and phenomenological responses would fit into some coherent system. It would have also helped the reader to follow all the details, if such an attempt to generate some common narrative would have been made.

We now rewrote the second half of the abstract (line 6-12), rewrote the discussion in its entirety (lines 326-450), and added a circuit schematic as an additional supplementary figure (Supplementary Fig. 8). In particular, we now end the discussion with the main conclusion, which is that local processing in the superficial layers is necessary for grid firing (line 445-450: *'While processing of afferent information by these functional cell types and recurrent circuits between grid cells have been the standard tenet of computational models for grid firing^{13,14}, our results are a critical experimental confirmation that key computations for grid accuracy are performed within the superficial layers. Finally, our results provide evidence that, under*

circumstances of foraging with abundant sensory cues, any local perturbations of position accuracy are corrected on a timescale in the order of milliseconds.’ Our data therefore provide the first experimental support for a large number of standard grid cell models, which have proposed that processing in local networks within the superficial layers is critical for grid patterns.

Major comments:

1. Fig. 2c – The cells are divided into different categories. How were the categories determined? This is not completely clear from the paper. Were they forming a continuum, or rather arbitrary divisions? How much are the results dependent on the thresholds used for categorization?

We used shuffling procedures to determine cutoffs (see Supplementary figures 2b and i). Most critical to our results is the classification of cells that were inhibited because the critical analysis was predominantly performed during the window of inhibition. Although there is somewhat of a continuum of these cells, Supplementary Fig. 2i shows a strongly skewed distribution with only few cells inhibited by less than 50 %. Moving the threshold over a range, e.g., from the 95th percentile (corresponding to ~35% inhibition) to the 90th percentile (corresponding to ~30% inhibition) only minimally changes the number of cells classified as inhibited (from 205 to 210). While changing the cutoff to the 99th percentile would result in a drastic decrease in the number of cells classified as inhibited, we believe that the 99th percentile threshold is too conservative. It would exclude a large fraction of cells that are inhibited by more than 50 % from being classified as inhibited. From the comment below, we gather that the reviewer is also concerned about us classifying too many cells as not responding. Choosing an extremely conservative threshold for inhibition would further exacerbate the concern.

To be more clear in our description of the procedures for determining cutoffs and to explicitly state that we used the 95th percentile of the baseline distribution, we rewrote the relevant section in the Methods (line 986-1027) and also rewrote the legend of the relevant supplementary figure (Supplementary Fig. 2).

2. Fig 2c - Some of the “No response” examples seem to contain a small inhibitory response that cannot be neglected, and is reflected in the summed population activity (red trace). This same issue is seen in Fig. 5c – while it is stated in the text that there is no response, clearly there is a small inhibitory response in many of the cells, which is reflected also in the population activity.

We agree that some responses are apparent, but that these responses are extremely weak such that the cells would not be appropriately included in any analysis of the much more strongly inhibited cells. However, to fully acknowledge that a cell that is classified as non-responsive might have a weak response, we are now careful in referring to this class of cells throughout the text and in figures and legends as the ‘No/weak response’ group (e.g., Fig. 2, 3, 5, 6, and 7).

In addition, we quantified any remaining responses in the no/weak response group and compared them to baseline and cells that were classified as inhibited. This analysis is shown below and indicates that the reduction in rate for the no/weak response group did not differ from baseline and was vastly different from the cells that were classified as inhibited (Kruskal-Wallis test followed by Dunn's multiple comparisons test, $n = 266$ (Baseline), 205 (Inhibition), 34

(no/weak response) cells from 7 mice, $H(3) = 349.6$, $p < 0.0001$). We note that the statistical difference here is expected because cells in the inhibition group were selected to be inhibited, and we would therefore opt to not include a trivial result in the manuscript. However, we nonetheless consider the large difference in effect sizes between the Inhibition and No/weak response group as an indication that our classification criteria capture a meaningful difference between the groups.

3. Fig. 2f – the clustering of interneurons should be done while taking into account both axes, and not only firing rate. For example, 4 of the gray diamonds (and some other cells there) seem to belong to the excitatory neuron cluster on the left, while some of the pink dots with a low firing rate (those with a peak-valley ratio below about 0.7) actually clearly belong to the interneuron cluster. Please re-cluster the cells and see how it affects the results. It seems that it will only strengthen the paper's findings.

As suggested, we reclustered the interneurons in Fig. 2f and are now reporting all the results based on the new clustering criteria. The reclassification did not alter any of our main results, which confirms that our results are robust for a range of classification criteria.

4. Fig. 4c – The authors seem to find a potential resonance effect at about 8 Hz. However they have not tried a lower frequency of stimulation (say at 4 or 6 Hz), so they should reserve their claim about 8 Hz, as it could have been that the stronger effect would have been found at an even lower frequency of stimulation.

We did not mean to imply that there was resonance. Rather, we observed population spikes in response to local stimulation at both 8 Hz and 12 Hz, and in the original manuscript, only performed limited analysis of theta oscillations in the 8 Hz stimulation condition because the light-elicited population spikes precluded that we could cleanly apply band-pass filters that distinguished between the endogenous oscillations and the evoked signals. At 12 Hz, the population spikes and theta oscillations also co-occurred, but separate analysis of the endogenous theta band was possible because there was no overlap with the stimulation frequency.

We now rephrased the text to not inadvertently suggest resonance effects, and the text now reads (line 203-217):

'For our analysis of theta oscillations, we had to consider that our optogenetic stimulation generated population spikes in the LFP signal, which were superimposed onto endogenous LFP and did not allow us to measure theta power (6-10 Hz) independently of stimulation effects for 8 Hz stimulation (Fig. 4a, b). For 12 Hz stimulation sessions, however, we could clearly distinguish two separate frequency bands – one at 12 Hz, arising due to the population spikes and another at theta frequencies – and were able to show that there was no detectable increase in the endogenous theta amplitude from population spikes (Fig. 4c). We next performed a linear regression between running speed and either the peak theta frequency (Fig. 4d) or average theta power (Fig. 4e) and found no difference between the slopes of baseline and stimulation sessions for both measurements for 12 Hz sessions. For 8 Hz sessions, population spike artifacts remained in the filtered signal (6-10 Hz) and increased the average theta power, while the power nonetheless remained modulated by running speed to the same extent as during baseline. Theta-related speed information was thus retained in the mEC throughout sessions with local mEC circuit perturbation.'

5. Suppl. Fig 4b – While the narrow HD-cells form a cluster, the wide-HD cells seem to form a continuum with unclassified cells (The blue and black dots are not separable as clusters). How does changing the criterion to a 99-percentile (for this a shuffling of 1000 will be required) change your results?

The shuffling was originally performed with 1000 iterations and a 95-percentile threshold, and we redid the analysis with the 99-percentile threshold. As expected, the total number of cells that were classified as broad HD cells decreased (from 58 to 38), but our results remained the same, as shown in the plot below. Because the criterion did not alter the results, we chose to retain the 95-percentile cutoff to be consistent with the classification for the other cell types. However, if the reviewer considers that adding the results with the 99-percentile threshold would substantially strengthen our finding, we would of course agree to add all relevant plots.

Figure X. (a) Scatter plots of the maximum HD score of all LII and LIII cells versus the 99-percentile of scores from shuffled data. All narrowly tuned head direction cells that were originally classified using a 95th percentile cutoff were also above the 99-percentile threshold. Of the broad HD cells (blue dots), fewer (38 rather than 58) were selected with the more stringent threshold. (b) Peri-stimulus time plot for all broad HD cells that were selected with the 99-percentile cutoff. The overall responses of the cells are similar to those observed with a 95-percentile cutoff (see Figure 6b). (c) Quantification of HD tuning of all inhibited broad HD cells (classified using a 99-percentile cutoff) shows similar results as those observed with a 95-percentile cutoff (One-way repeated measures ANOVA followed by Tukey's multiple comparisons test, $n = 18$ cells from 7 mice, $F(1.515, 25.76) = 9.958$, $p = 0.0014$).

6. The Discussion part does not attempt to try to connect the different phenomena together, and provide some explanation as to why these phenomena occur the way they do in the different subsets of cells. Furthermore, the discussion trying to connect the phenomena to the thalamus is very weak and farfetched, and should be removed.

As suggested, we now entirely rewrote the discussion (lines 326-450), where we mention standard grid cell models as a framework for our observed results and state that grid cells were perturbed to a different extent than would be expected if they merely processed afferent/speed information. We also added a schematic to summarize our findings (Supplementary Fig. 8) and removed the part of the discussion that connected our observations to the thalamus.

Minor comments:

1. It is better to change the title to include also the speed component: “OPTOGENETICALLY INDUCED FEEDBACK INHIBITION OF MEDIAL ENTORHINAL CORTEX SUPERFICIAL LAYERS TRANSIENTLY PERTURBS GRID ACCURACY BUT NOT HEAD DIRECTION and speed TUNING”

We changed the title as suggested and, in response to a comment by reviewer #2, also edited the abstract to address the concern that speed cells are not sufficiently mentioned.

2. Fig. 1e – The distribution does not seem Gaussian, so it is better to use a non-parametric t-test (Same comment for Suppl. Fig. 1c,d).

Changed as suggested, and the outcome remained unchanged from the original manuscript.

3. Fig. 1g – the authors should elaborate a bit more, for the sake of clarity, how the image explains their claim?

The figure legend for panel g is now expanded and now reads: *‘Consistent with previous reports on the projections of LIIP cells to the hippocampus, we observed synapses from LIIP cells in CA1. We did not observe synapses in the DG-CA3 region, where LIIS cells are known to terminate, which further confirms that the Wfs1 line is selective for LIIP cells.’*

4. Fig. 2c – For readability, please sort the cells in the raster plots according to the estimated latencies of the response.

Cells in the ‘Only excitation’ and ‘Excitation & inhibition’ groups were already sorted by the latency, which may not have been apparent because of the size allotted to individual cells. We therefore now expanded these panels in the vertical direction. In addition, the ‘Only Inhibition’ group is now sorted by the onset of inhibition.

5. Fig. 2f – ratios should be plotted on a logarithmic axis. For example, the peak-valley ratio in the y-axis of this panel should be logarithmic. Also the scale (and calculation of errors) in the y-axis of fig. 2g should be logarithmic for the same reason.

Now replotted on a logarithmic scale.

6. Fig. 2f – the color legend in panel e does not completely match the colors of the points. There is no gray in the legend, so it is not completely clear if the gray means only inhibition or no response.

We now changed the color from white to gray in 2e such that it is clear that ‘Only inhibition’ corresponds to gray. This was also the intent in the original figure but we realize that we were inconsistent in using either white or gray for the ‘Only inhibition’ group. The dark blue is changed to light blue to clearly distinguish blue from black.

7. Fig. 3 – It is not clear if any of the speed cells shown in this figure were classified as interneurons

As in the publication that originally identified speed cells in mEC (Kropff et al., 2015) and in a recent publication that identified most speed cells as fast spiking (Ye et al., 2018), we included interneurons. We now explicitly report the fraction of speed cells that are interneurons (i.e., 15 of 47, 31.9 %) in the main text (line 173) and figure legend (Figure 3, line 675). In addition, we provide an updated detailed breakdown of cell type, speed score, and firing rate in Supplementary Figs. 4f and g.

8. Fig. 4d and 4e – The graphs for 8 Hz should be shown too.

We added these panels, but emphasize that the LFP data shown for 8 Hz is at least partly contaminated by population spikes arising due to the optogenetic stimulation. This is explicitly mentioned in the text (also see response to major comment #4).

9. Supp Fig.1a – "the damage observed in the sections is caused by postmortem tetrode extraction". Could the implantation itself not be a reason too?

We agree that a portion of the damage could also be done at the time of implantation and therefore added a corresponding comment to the legend of Supplementary Fig. 1a: *‘While the deeper damage along tracks is from advancing tetrodes and the optical fiber, the tears in the tissue were observed to be caused by the postmortem extraction of the implant.’*

10. Suppl. Fig. 2b – The measure of Peak is very noisy, and highly depends on the level of smoothing. How does this classification depend on the smoothing of the signal?

We selected 1 ms bins for the calculation of the PTSH because we reasoned that 1 ms would already be a relatively coarse resolution and sufficiently smooth the data to result in reliable peak detection. To further substantiate this point, we now calculated the PSTH across a range of resolutions (0.1 ms, 1 ms and 2 ms). The robustness of the classification when using either a 1 ms or 2 ms bin size is evident by calculating the 99th (green vertical line), 95th (red vertical line)

and 90th (black vertical line) percentile cutoff and by then determining the number of cells that were classified as excited or inhibited with each cutoff (*numbers above each vertical line*). Variability increases with the smallest bin size (0.1 ms) for excitation, but not inhibition. Classification thus appears robust with bin sizes that smooth over at least 1 ms.

Figure Y. The variability in the firing rate during baseline (40 to 0 ms before light onset) was measured by calculating the PSTH using 0.1 ms (*top*), 1 ms (*middle*) and 2 ms (*bottom*) wide bins (*grey bars*), and by selecting the bin with the highest normalized rate (*left column*) or the 10 ms window with the lowest normalized rate (*right column*) over 40 ms before (*grey*) and after (*blue, yellow*) light stimulation. As can be seen from the plots, the variability during baseline increased extensively with a bin size of 0.1 ms (and thus less smoothing) for the excitation classification (*bottom left*), for which the firing rate within a single bin was used. Because smoothing with a 10 ms wide sliding window was used for measuring inhibition, there was no substantial difference in variability from varying the bin size. Bin size for calculating the PSTH therefore had a lesser effect for inhibition than for excitation, but effects for excitation were also minimal with bin sizes ≥ 1 ms.

11. Suppl. Fig. 2j is important. It should be moved to a main figure.

We agree that showing that the latency of inhibition is longer than the latency of excitation is important and moved the panel to Fig. 2i.

12. Suppl. Fig. 4b – please make the figures square (that is “axis equal”). Also, please show a histogram of the HD score, to emphasize the bi-modality. In addition, it is not clear what are black dots doing above the diagonal? I thought this figure was used for the determination of the colors of the dots.

The panels are now square and a histogram of the HD score is added. In the plots, we show scores for a single session, but cells were classified as belonging to a particular class with the more conservative criterion that they had to exceed threshold in at least 2 of 3 sessions. Grey dots above the line are thus cells that were above threshold in only one session. This is now explicitly stated in the legend (*‘Thus, grey dots above the line are cells that were above the threshold in only one session.’*).

13. Supplementary Figure 5 – “The phase angle preference was preserved across sessions.” While generally true, the “Baseline 2” map does not show a very good recovery of tuning of the cells.

Please note that the phase angle is ordered by baseline 1 and that the differences that emerge are expected based on variability across sessions. This is shown here by also ordering the same data by the stimulation session and by the baseline 2 session. A similar amount of variability is observed for each condition to which the ordering was not applied. If these different types of displays are not considered redundant, we would of course follow the advice to include them in the manuscript.

REFERENCES

Bonnevie, T., Dunn, B., Fyhn, M., Hafting, T., Derdikman, D., Kubie, J.L., Roudi, Y., Moser, E.I., and Moser, M.-B. (2013). Grid cells require excitatory drive from the hippocampus. *Nat. Neurosci.* 16, 309–317.

Brandon, M.P., Bogaard, A.R., Libby, C.P., Connerney, M.A., Gupta, K., and Hasselmo, M.E. (2011). Reduction of theta rhythm dissociates grid cell spatial periodicity from directional tuning. *Science* 332, 595–599.

Buetfering, C., Allen, K., and Monyer, H. (2014). Parvalbumin interneurons provide grid cell-driven recurrent inhibition in the medial entorhinal cortex. *Nat. Neurosci.* 17, 710–718.

Couey, J.J., Witoelar, A., Zhang, S.-J., Zheng, K., Ye, J., Dunn, B., Czajkowski, R., Moser, M.-B.,

- Moser, E.I., Roudi, Y., et al. (2013). Recurrent inhibitory circuitry as a mechanism for grid formation. *Nat. Neurosci.* *16*, 318–324.
- Dunn, B., Mørreanet, M., and Roudi, Y. (2015). Correlations and Functional Connections in a Population of Grid Cells. *PLOS Comput. Biol.* *11*, e1004052.
- Ego-Stengel, V., and Wilson, M.A. (2010). Disruption of ripple-associated hippocampal activity during rest impairs spatial learning in the rat. *Hippocampus* *20*, 1–10.
- Fuchs, E.C., Neitz, A., Pinna, R., Melzer, S., Caputi, A., and Monyer, H. (2016). Local and Distant Input Controlling Excitation in Layer II of the Medial Entorhinal Cortex. *Neuron* *89*, 194–208.
- Gil, M., Ancau, M., Schlesiger, M.I., Neitz, A., Allen, K., De Marco, R.J., and Monyer, H. (2018). Impaired path integration in mice with disrupted grid cell firing. *Nat. Neurosci.* *21*, 81–91.
- Hinman, J.R., Brandon, M.P., Climer, J.R., Chapman, G.W., and Hasselmo, M.E. (2016). Multiple Running Speed Signals in Medial Entorhinal Cortex. *Neuron* *91*, 666–679.
- Jadhav, S.P., Kemere, C., German, P.W., and Frank, L.M. (2012). Awake Hippocampal Sharp-Wave Ripples Support Spatial Memory. *Science* (80-.). *336*, 1454–1458.
- Justus, D., Dalügge, D., Bothe, S., Fuhrmann, F., Hannes, C., Kaneko, H., Friedrichs, D., Sosulina, L., Schwarz, I., Elliott, D.A., et al. (2016). Glutamatergic synaptic integration of locomotion speed via septoentorhinal projections. *Nat. Neurosci.* *20*, 16–19.
- Kanter, B.R., Lykken, C.M., Avesar, D., Weible, A., Dickinson, J., Dunn, B., Borgesius, N.Z., Roudi, Y., and Kentros, C.G. (2017). A Novel Mechanism for the Grid-to-Place Cell Transformation Revealed by Transgenic Depolarization of Medial Entorhinal Cortex Layer II. *Neuron* *93*, 1480–1492.e6.
- Koenig, J., Linder, A.N., Leutgeb, J.K., and Leutgeb, S. (2011). The spatial periodicity of grid cells is not sustained during reduced theta oscillations. *Science* *332*, 592–595.
- Kropff, E., Carmichael, J.E., Moser, M.-B., and Moser, E.I. (2015). Speed cells in the medial entorhinal cortex. *Nature* *523*, 419–424.
- Mallory, C.S., Hardcastle, K., Bant, J.S., and Giocomo, L.M. (2018). Grid scale drives the scale and long-term stability of place maps. *Nat. Neurosci.* *21*, 270–282.
- Miao, C., Cao, Q., Moser, M.-B., and Moser, E.I. (2017). Parvalbumin and Somatostatin Interneurons Control Different Space-Coding Networks in the Medial Entorhinal Cortex. *Cell* *171*, 507–521.e17.
- Moser, E.I., Moser, M.-B., Lipa, P., Newton, M., Houston, F.P., Barnes, C.A., and McNaughton, B.L. (2005). A test of the reverberatory activity hypothesis for hippocampal “place” cells. *Neuroscience* *130*, 519–526.
- Sun, C., Kitamura, T., Yamamoto, J., Martin, J., Pignatelli, M., Kitch, L.J., Schnitzer, M.J., and Tonegawa, S. (2015). Distinct speed dependence of entorhinal island and ocean cells, including respective grid cells. *Proc. Natl. Acad. Sci. U. S. A.* *112*, 9466–9471.
- Tocker, G., Barak, O., and Derdikman, D. (2015). Grid cells correlation structure suggests organized feedforward projections into superficial layers of the medial entorhinal cortex.

Hippocampus 25, 1599–1613.

Winter, S.S., Clark, B.J., and Taube, J.S. (2015). Spatial navigation. Disruption of the head direction cell network impairs the parahippocampal grid cell signal. *Science* 347, 870–874.

Winterer, J., Maier, N., Wozny, C., Beed, P., Breustedt, J., Evangelista, R., Peng, Y., D’Albis, T., Kempter, R., and Schmitz, D. (2017). Excitatory Microcircuits within Superficial Layers of the Medial Entorhinal Cortex. *Cell Rep.* 19, 1110–1116.

Ye, J., Witter, M.P., Moser, M.-B., and Moser, E.I. (2018). Entorhinal fast-spiking speed cells project to the hippocampus. *Proc. Natl. Acad. Sci. U. S. A.* 115, E1627–E1636.

Zugaro, M.B., Monconduit, L., and Buzsáki, G. (2005). Spike phase precession persists after transient intrahippocampal perturbation. *Nat. Neurosci.* 8, 67–71.

Reviewers' comments:

Reviewer #1 (Remarks to the Author):

I thank the authors for their careful reply to my and the other referee's comments. My opinion has remained the same: while the data are well analyzed and the paper well written, the scope of this study remains somehow limited. The only take-home message seems to be that briefly activating MEC LII pyramidal neurons transiently perturbed the activity of the network while preserving the firing rate of finely tuned HD neurons and the information per spike of all other cell types. The only unexpected change is slight reduction in grid cell accuracy, reported in Figures 7i and 7k.

In their rebuttal, the authors claim that "It is our work that is the first [...] to demonstrate a role of the superficial layers in determining the location accuracy of grid patterns." I don't think this work convincingly demonstrate the role of the superficial layers in grid cell accuracy, although the hypothesis is likely true. Again, grid cell accuracy is reduced during the post-excitation inhibition, a state in which many of the physiological factors of the neurons are recovering from the strong and synchronous activation of LIIP neurons. A close inspection of figure 7f demonstrates, to me at least, that despite a massive effect on the network dynamics, it is instead quite remarkable that these 13 spikes fall so accurately in the grid fields (2/13 were convincingly out of a field, two were on the edge). Yet, the analysis of the normalized distance to the closest grid field centroid yields a significant result. However, this analysis seems misleading: the authors computed for each spike the distance from the current position to the closest centroid for the N number of spikes that were fired in the I state and randomly chosen N number of spikes in the Pre state. These values were normalized by the average distance of all spikes in the Pre condition. In other words, the normalization is made with control data, leading to a remarkable narrow distribution of average distance of control data around 0 in Fig 7k. The correct way would be to use baseline data or, even better, to normalize by a less arbitrary distance such as the grid spacing in baseline condition. The measure of accuracy, which is another way to analyze this phenomenon, also is also reduced during the inhibition (Fig 7i), but, again, these few spikes that fall out of field could result from non-specific physiological response, such as post-inhibitory rebound, and it is not very surprising that the accuracy is affected. This latter possibility is actually supported by the correlation between normalized average distance (if the measure is valid) and the I/Pre rate ratio: the stronger the cell is inhibited, the more rectifying currents will control its firing, certainly in a non-specific (i.e. non-spatial) manner for the first spikes of the post-inhibitory rebound.

Unless the authors can show, using this unique opportunity to label such a specific cell class, that LIIP cells play a specific role in the circuit, this manuscript lack a clear and novel message that will advance our knowledge of the mechanisms underlying grid cell generation in the MEC.

Reviewer #2 (Remarks to the Author):

This is a revised manuscript in which the authors have responded well to most of my previous comments. There are, however, still a couple of issues to address.

The authors contend that the speed signal is 'not substantially further processed within the mEC' as the primary speed signal is projected there presumably from the septal area. It was not clear from the paper how the data would look if the speed signal was processed further. What parameters of the speed signal would change?

What is meant by the term 'cellular resonance'? Please define more clearly.

Reviewer #3 (Remarks to the Author):

The authors have nicely addressed all my comments.

On a second reading of the paper and the rebuttal, I reckon that while the message of the paper is rather subtle, it is an important one. The loss of accuracy of grid cells due to the local network perturbation is a major constraint to be taken into account in future theories of grid cell formation.

It would have been interesting to check whether a similar analysis on an entorhinal stellate-cells perturbation experiment (Such as that of Kanter et al., 2017) would have reached a different conclusion concerning grid cell field accuracy during inhibition. Perhaps if the authors have such data they could check that.

I have two requests concerning display of figures:

- a) Figure X from the rebuttal is important, and thus should be added to the supplementary figures.
- b) Supplementary figure 8 is important for comprehension of the Discussion section, and thus it should be added as a sub-panel into one of the figures in the main text (although it is somewhat redundant with figure 1a).

RESPONSE TO RE-REVIEW, NCOMMS-17-28893B

We thank the three reviewers for their thorough re-evaluation of our manuscript and apologize that there are remaining points that we have not been able to fully address in the previous revision. These remaining comments are now addressed with additional analysis, as described in our point-by-point responses. We retained reviewer comments in black font and added our replies in blue font. In addition, for ease of identifying changes to the main text, we marked all edits in the revised manuscript with red font.

Reviewers' comments:

Reviewer #1 (Remarks to the Author):

I thank the authors for their careful reply to my and the other referee's comments. My opinion has remained the same: while the data are well analyzed and the paper well written, the scope of this study remains somehow limited. The only take-home message seems to be that briefly activating MEC LII pyramidal neurons transiently perturbed the activity of the network while preserving the firing rate of finely tuned HD neurons and the information per spike of all other cell types. The only unexpected change is slight reduction in grid cell accuracy, reported in Figures 7i and 7k.

In their rebuttal, the authors claim that "It is our work that is the first [...] to demonstrate a role of the superficial layers in determining the location accuracy of grid patterns." I don't think this work convincingly demonstrate the role of the superficial layers in grid cell accuracy, although the hypothesis is likely true. Again, grid cell accuracy is reduced during the post-excitation inhibition, a state in which many of the physiological factors of the neurons are recovering from the strong and synchronous activation of LIIP neurons. A close inspection of figure 7f demonstrates, to me at least, that despite a massive effect on the network dynamics, it is instead quite remarkable that these 13 spikes fall so accurately in the grid fields (2/13 were convincingly out of a field, two were on the edge). Yet, the analysis of the normalized distance to the closest grid field centroid yields a significant result. However, this analysis seems misleading: the authors computed for each spike the distance from the current position to the closest centroid for the N number of spikes that were fired in the I state and randomly chosen N number of spikes in the Pre state. These values were normalized by the average distance of all spikes in the Pre condition. In other words, the normalization is made with control data, leading to a remarkable narrow distribution of average distance of control data around 0 in Fig 7k. The correct way would be to use baseline data or, even better, to normalize by a less arbitrary distance such as the grid spacing in baseline condition.

The measure of accuracy, which is another way to analyze this phenomenon, also is also reduced during the inhibition (Fig 7i), but, again, these few spikes that fall out of field could result from non-specific physiological response, such as post-inhibitory rebound, and it is not very surprising that the accuracy is affected. This latter possibility is actually supported by the correlation between normalized average distance (if the measure is valid) and the I/Pre rate ratio: the stronger the cell is inhibited, the more rectifying currents will control its firing, certainly in a non-specific (i.e. non-spatial) manner for the first spikes of the post-inhibitory rebound.

Unless the authors can show, using this unique opportunity to label such a specific cell class, that LIIP cells play a specific role in the circuit, this manuscript lack a clear and novel message that will advance our knowledge of the mechanisms underlying grid cell generation in the MEC.

We completely agree with the summary of the results by the reviewer, which is that there was a massive effect on network dynamics, which surprisingly yielded only a modest reduction in grid accuracy and no effects on HD tuning and speed tuning. Based on prior theoretical work on grid cells, we approached our study with the expectation that there would be a pervasive effect on most functional cell types in the MEC from a manipulation that inhibits > 80 % of the cells in the superficial layers. However, the study by Kanter et al. and our own initial cursory analysis of the data (analyzing the entire recording session, as shown in Fig. 7b) left us with the initial conclusion that local circuits are not essential for grid firing. As stated by the reviewer, network models predicted that the correct answer is more likely that local circuit computations are necessary for grid generation. We therefore proceeded to perform the careful detailed analysis that is described in the remainder of Fig. 7 and would like to thank the reviewer for repeatedly commending us on the thoroughness of our analysis procedures, such that we did not miss the effect on grid cells even though it was more subtle than expected.

While our finding of a partial effect on grid cells by massive local silencing is certainly below the expectation of some grid cell models, we see our results as pivotal in providing a counterpoint to the results by Kanter et al. While Kanter et al. find that rate coding by grid cells is altered, we show that local manipulations of the superficial layers can have effects on location accuracy. However, the finding that the effect size is lower than anticipated is also important in contributing to a conceptual recalibration of the expected balance between intrinsic and afferent circuitry. A take home message of our results is thus that there is at least a minor local circuit contribution, but that afferent inputs are nonetheless critical and can rapidly correct for errors, consistent with prior work on major effects on grid cells by manipulating inputs from the septal area, the HD system, and the hippocampus.

With regard to the final point by the reviewer on identifying a more specific role of LIIP cells, we would like to point out that our study hints at a very real limitation of cell-type selective manipulations. The capability of being able to label a cell class does not imply that the cell class can be manipulated in isolation. Rather than ignoring the fact that any optogenetic or chemogenetic manipulation has off-target effects, we therefore measured the effects on all cells in the local circuit and made use of the strong local connectivity of LIIP cells to massively inhibit the superficial layers, which led to the finding of a modestly decreased accuracy of grid firing.

Central to our claim is of course that our analysis in Fig.7k is not misleading, and we therefore closely followed the advice of the reviewer to reconsider the normalization procedure. We thank the reviewer for suggesting to consider grid spacing as the best option for normalization. Using this option, we of course find much higher baseline variability, but nonetheless continue to see that the spikes during the period of inhibition are farther from the grid centers than the corresponding control spikes (One-way repeated measures ANOVA followed by Tukey's multiple comparisons test, $n = 39$ cells from 3 mice, $F(1.707, 64.86) = 137$, $p < 0.0001$). A detailed inspection of the new panel j reveals that this effect is seen in almost all grid cells [30 of 39 cells for which spikes during inhibition (green dots) are above spikes during pre (red dots)]. Furthermore, it is important to note that the effect size for many cells is substantial given that the baseline distance for an intact grid cell is ~ 0.15 and that the randomized distance is ~ 0.25 . Note that a randomized value of 0.25 is expected because the minimum distance is 0 (for spikes at a grid node) and the maximum distance is 0.5 (for spikes in between two nodes; if more than 0.5 from one node, the spike would be less than 0.5 from another node). The average distance for spikes that are evenly distributed between 0 and 0.5 is thus 0.25. The average distance for spikes during inhibition is ~ 0.18 , which is a substantial increase from control spikes when considering that the possible range is between intact grid firing and randomized firing.

To absolutely make sure that the results of our distance analysis are not biased by any normalization procedure, we also checked the average distance of spikes from the grid centers without normalization (top left, One-way repeated measures ANOVA followed by Tukey's multiple comparisons test, $n = 39$ cells from 3 mice, $F(1.558, 59.19) = 140.4$, $p < 0.0001$), with normalization to distance during the baseline session (top right, One-way repeated measures ANOVA followed by Tukey's multiple comparisons test, $n = 39$ cells from 3 mice, $F(1.659, 63.05) = 122.3$, $p < 0.0001$; One-sample t-tests compared to 1, Pre: $t(38) = 0.2358$, $p = 0.8148$, I: $t(38) = 2.922$, $p = 0.0058$, Path coordinates: $t(38) = 14.29$, $p < 0.0001$), and with normalization to the radius of grid cells (bottom right, One-way repeated measures ANOVA followed by Tukey's multiple comparisons test, $n = 39$ cells from 3 mice, $F(1.615, 61.36) = 121.5$, $p < 0.0001$). For all possibilities, including the normalization to grid spacing that is now shown in the manuscript (bottom left, One-way repeated measures ANOVA followed by Tukey's multiple comparisons test, $n = 39$ cells from 3 mice, $F(1.707, 64.86) = 137$, $p < 0.0001$), we found a clear and significant increase in distance during the period of inhibition.

If requested, we could therefore show any of the panels above in the manuscript. Furthermore, to make the measurements that are based on distance from grid nodes and on % spikes in grid fields better comparable, we also inverted the axis for Fig. 7 e-h and now replaced the decrease of in-field spikes with the increase of out-of-field spikes. Although this is numerically and statistically identical to the original manuscript, it has the advantage that both measurements now show an increased position error in the same direction. We thought that this would increase the clarity of the presentation in Fig. 7, but could of course revert to the previous presentation, if requested. In addition, to better show the increased out-of-field spiking, we replaced the example cell in Fig. 7 with one that fires a total of 69 spikes during the period of inhibition. The sample from the original figure is still shown but now in Supplementary Fig. 7 (Cell 7).

The final point to address is whether our results could be explained by rebound spiking. While we would argue that the classical definition of rebound spiking (e.g., spiking when a cells is rapidly stepped from a hyperpolarized to a more depolarized membrane potential) does not apply, the more relevant question is whether the spikes during the period of inhibition are purely non-physiological responses unrelated to excitation of cells by afferent or local input. Our strongest argument to exclude this possibility is that the

randomly generated spikes in response to rebound from inhibition would result in a disruption of all firing properties – grid, speed, and HD – to near-random levels. It would of course be conceivable that only grid cells have the channel composition to generate rebound spiking in response to LIIP cell elicited inhibition, but this notion is not consistent with the selective effect on grid accuracy as opposed to head direction tuning of grid-by-HD cells. To further quantify the difference between effects on grid but not HD tuning, we added panel j to Supplementary Fig. 7, which shows that not only HD length (panel i, unchanged from previous version) but also HD angle remains intact (new panel j) while grid firing is disrupted (panel h).

Reviewer #2 (Remarks to the Author):

This is a revised manuscript in which the authors have responded well to most of my previous comments. There are, however, still a couple of issues to address.

The authors contend that the speed signal is ‘not substantially further processed within the mEC’ as the primary speed signal is projected there presumably from the septal area. It was not clear from the paper how the data would look if the speed signal was processed further. What parameters of the speed signal would change?

If local processing were required to further sharpen or amplify the speed signal, we would have expected that the speed scores decreased during the period of inhibition beyond the level that could be predicted from the rate reduction. To better convey this expectation in the discussion, we modified the corresponding sentence to now read, *‘The observation that speed coding was not compromised beyond the predicted level suggests that local processing is not required to further amplify or sharpen the rate-coded speed signal in mEC (line 384-386).’*

What is meant by the term ‘cellular resonance’? Please define more clearly.

We used the term ‘resonance properties of a cell’ to imply that different anatomical cell types differ in their degree of resonance. To now define more clearly how different excitatory cells differ we replaced ‘resonance’ with ‘ion channel composition’ such that the sentence now reads, *‘The limited effects of local circuit manipulations on grid cells therefore raise the possibility that dendritic processing or ion channel composition of a cell predominantly contribute to grid generation, and that grid firing may thus selectively emerge in a particular morphological cell type (line 43-46).’*

Reviewer #3 (Remarks to the Author):

The authors have nicely addressed all my comments.

On a second reading of the paper and the rebuttal, I reckon that while the message of the paper is rather subtle, it is an important one. The loss of accuracy of grid cells due to the local network perturbation is a major constraint to be taken into account in future theories of grid cell formation.

Thank you for emphasizing the importance of the message and that a subtle result can be an important one.

It would have been interesting to check whether a similar analysis on an entorhinal stellate-cells perturbation experiment (Such as that of Kanter et al., 2017) would have reached a different conclusion concerning grid cell field accuracy during inhibition. Perhaps if the authors have such data they could check that.

We unfortunately do not have these data but are planning to perform experiments along the lines suggested by the reviewer. We envision that these follow-up experiments are best done as part of a broader research program that also establishes the consequences of layer II stellate cell manipulations on layer V cells, which are the main intraentorhinal targets of stellate cells.

I have two requests concerning display of figures:

a) Figure X from the rebuttal is important, and thus should be added to the supplementary figures.

As suggested, the figure from the rebuttal is now added to Supplementary Fig. 4 as panels h-j. The main text [*‘These results were also reproduced with a more conservative, 99th percentile threshold to classify broad HD cells (Supplementary Fig. 4h-j), line 260-262’*] and figure legend are updated accordingly.

b) Supplementary figure 8 is important for comprehension of the Discussion section, and thus it should be added as a sub-panel into one of the figures in the main text (although it is somewhat redundant with figure 1a).

We moved the figure to the main text. Because the figure is most relevant to the discussion, we reasoned that it best fits with the flow of the text as an additional main figure (now Fig. 8).

REVIEWERS' COMMENTS:

Reviewer #1 (Remarks to the Author):

I thank the authors for carefully addressing my last concerns. The additional analyses are compelling and nicely show that, indeed, the manipulation seems to be specific to grid cell accuracy. I thus have no further concern.

Figure 7 does not need to include the analyses presented in the response.